# Solving Multiobjective Combinatorial Optimization via Learn to Improve Method

## Abstract

Recently, deep reinforcement learning (DRL) has been prevailing for solving multiobjective combinatorial optimization problems (MOCOPs). Most DRL methods are based on the "Learn to Construct" paradigm, where the trained model(s) can directly generate a set of approximate Pareto optimal solutions. However, these methods still suffer from insufficient proximity and poor diversity towards the true Pareto front. In this paper, following the "Learn to Improve" (L2I) paradigm, we propose weight-related policy network (WRPN), a learning-based improvement method for solving MOCOPs. WRPN is incorporated into multiobjective evolutionary algorithm (MOEA) frameworks to effectively guide the search direction. A shared baseline for proximal policy optimization is presented to reduce variance in model training. A quality enhancement mechanism is designed to further improve the Pareto set in model inference. Computational experiments conducted on two classic MOCOPs, i.e., multiobjective traveling salesman problem and multiobjective vehicle routing problem, indicate that our method achieves state-of-the-art results. Notably, our WRPN module can be easily integrated into various MOEA frameworks such as NSGA-II, MOEA/D and MOGLS.

## 1 Introduction

Multiobjective combinatorial optimization problems (MOCOPs) (Ehrgott and Gandibleux, 2000) have wide applications in various fields, such as communication routing, investment planning, vehicle routing, logistics scheduling, etc. Solving such kind of problems requires taking into account different roles' preferences corresponding to different objectives, which may often conflict with each other. In principle, the goal of MOCOPs is to find the best compromise solutions (known as Pareto optimal solutions) rather than a single optimal solution. The decision maker can eventually choose a particular Pareto optimal solution according to his knowledge for practical usage.

MOCOPs have been extensively studied in computational intelligence communities in past decades. Seeking a set of Pareto optimal solutions for an MOCOP is extremely challenging, even its scalarized single-objective subproblem is generally NP-hard. In real-world applications, heuristic methods, mostly based on evolutionary algorithms, are commonly introduced to cope with MOCOPs. They can generate a set of approximately efficient solutions in reasonable time. However, traditional heuristics methods adopt simple rules or complex operations relied on experts' experience and knowledge for a specific problem, which may be limited to providing high-quality solutions for general MOCOPs.

Over the past few years, neural learning methods, especially deep reinforcement learning (DRL) methods, have made great achievements in solving single-objective combinatorial optimization problems (COPs) (Bengio et al., 2021; Mazyavkina et al., 2021; Wang and Tang, 2021). By capturing implicit patterns from a large number of problem instances, these methods can obtain better solutions than traditional heuristic methods in many scenarios.

In more recent years, there are several attempts trying to tackle MOCOPs via DRL. Most approaches (Li et al., 2020; Zhang et al., 2021; Lin et al., 2022; Zhang et al., 2022) follow the "Learn to Construct" (L2C) paradigm by using end-to-end learning models. Roughly speaking, they first decompose an MOCOP into multiple subproblems by different weight vectors, and then employ well-trained models to rapidly construct Pareto optimal solutions in the inference stage.

Despite some success of L2C methods has been achieved for MOCOPs, they still have their downsides. On the one hand, L2C methods heavily rely on the quality of end-to-end models for solving decomposed subproblems. Till now, even the state-of-the-art models (Kool et al., 2019; Kwon et al., 2020) still have substantial solution gaps for solving single-objective COPs, thereby resulting in the convergence (or proximity) problem. On the other hand, L2C methods directly construct a solution for a corresponding subproblem. It brings about the diversity problem due to the inadequate search for potential Pareto solutions in the neighborhood of each subproblem.

In order to address the above issues, we propose weight-related policy network (WRPN) based on another learning paradiam, named "Learn to Improve" (L2I) for solving MOCOPs. WRPN is embedded into typical multiobjective evolutionary algorithms (MOEAs) to perform potential improving operations in parallel for individual solutions in the population. It iteratively updates the current population and gradually approximates to the Pareto optimal solutions.

The contributions of our work can be summarized as follows.

- We propose WRPN, which is the first generic L2I method for MOCOPs. It can automatically provide efficient improvement operators for a batch of individual solutions, guided by a policy network that is designed to simultaneously extract the weight and solution features.

- We design a shared baseline, which is computed by a group of heterogeneous offsprings generated through evolution to realize low variance and mitigate the issue of local optima.

- We devise a quality enhancement mechanism. Based on instance augmentation techniques, it further utilizes the external population of MOEAs and the Pareto dominance of solutions, thereby improving the proximity and diversity of the Pareto set.

- We show that WRPN outperforms the existing state-of-the-art methods on classic MOCOPs. It is even superior to the excellent LKH solver (Tinós et al., 2018) under the weighted-sum decomposition for multiobjective traveling salesman problem (MOTSP). Notably, our WRPN module can be easily integrated into various MOEA approaches such as NSGA-II (Deb et al., 2002), MOEA/D (Zhang and Li, 2007) and MOGLS (Jaszkiewicz, 2002).

## 2 RELATED WORKS

**Exact and Heuristics Methods for MOCOPs.** Exact (Florios and Mavrotas, 2014) and heuristic (Herzel et al., 2021) algorithms are two groups of methods to solve MOCOPs in past decades. The former can find all the Pareto-optimal solutions for only very small-scale problems, while the latter, commonly used in practical applications, can find the approximate Pareto-optimal solutions within reasonable time. Multiobjective evolutionary algorithms are typical representatives of heuristic algorithms, including NSGA-II (Deb et al., 2002), MOEA/D (Zhang and Li, 2007), MOGLS (Jaszkiewicz, 2002), PLS (Angel et al., 2004), and PPLS/D-C (Shi et al., 2022).

**DRL methods for COPs.** In literature, some end-to-end DRL construction methods are developed for solving single-objective COPs. The pioneering works (Bello et al., 2017; Nazari et al., 2018) train a pointer network to construct a near-optimal solution for COPs. Kool et al. (2019) propose an Attention Model (AM) based on the Transformer architecture. A representative work is policy optimization with multiple optima (POMO) (Kwon et al., 2020), which exploits the symmetry of solutions to further improve the performance of end-to-end models. Distinguished from construction methods, improvement methods, another important class of DRL methods, iteratively improve the current solution, assisted by learning techniques. They generally achieve superior results compared with construction methods although longer running time may be taken. Typical works include Wu et al. (2021); Chen and Tian (2019); Lu et al. (2019); Ma et al. (2021) for vehicle routing problems.

**DRL methods for MOCOPs.** There are relatively few works using DRL to solve MOCOPs. Most of them are construction methods based on decomposition (Zhang and Li, 2007). Their basic idea is to decompose MOCOPs into multiple subproblems according to prescribed weight vectors, and then train a single model or multiple models to solve these subproblems. For example, Li et al. (2020); Zhang et al. (2021) train multiple models collaboratively through a transfer learning strategy. Preference-conditioned multi-objective combinatorial optimization (PMOCO) (Lin et al., 2022) trains a hypernetwork-based model, which can generate the decoder parameters according to weight vectors for solving subproblems. Meta-Learning-based DRL (MLDRL) (Zhang et al., 2022) first trains a

meta-model and then quickly fine-tunes the meta-model based on weight vectors to solve subproblems. To our best knowledge, PMOCO and MLDRL are two competitive DRL methods for MOCOPs.

## 3 PRELIMINARIES

**MOCOP.** The definition of an MOCOP can be described by $\min\limits_{x \in X} F(x) = (f_1(x), \cdots, f_M(x))$, where $X \in R^N$ is the feasible domain of $N$ decision variables, $F(x)$ is an $M$-dimensional objective vector and $f_i(x)$ represents the $i$-th objective function. Since the objectives are usually in conflict with each other, a set of trade-off solutions is sought. The concept of Pareto optimality is introduced.

**Definition 1 (Pareto dominance).** Let $u, v \in X$, $u$ is said to dominate $v$, i.e., $u \prec v$, if: 1) $\forall i \in \{1, \cdots, M\}, f_i(u) \leq f_i(v)$, and 2) $\exists j \in \{1, \cdots, M\}, f_j(u) < f_j(v)$.

**Definition 2 (Pareto optimality).** A solution $x^* \in X$ is called a Pareto optimal solution if $x^*$ is not dominated by any other solutions, i.e., $\nexists x' \in X : x' \prec x^*$. All the Pareto optimal solutions can be defined as $P := \{x^* \in X | \nexists x' \in X : x' \prec x^*\}$, which is the **Pareto set** (PS). All the Pareto optimal objective vectors constitute the **Pareto front** (PF).

**Utility Function.** A utility (or aggregated) function can map each point in the objective space into a scalar according to an $M$-dimensional weight vector satisfying $\sum_{i=1}^{M} \lambda_i = 1$ and $\lambda_i \geq 0$. Weighted-Sum (WS) and Weighted-Tchebycheff are commonly used utility functions (Miettinen, 2012). As the simplest representative, WS can be defined by $\min\limits_{x \in X} f(x|\lambda) = \sum_{i=1}^{M} \lambda_i f_i(x)$,

**Key Components of MOEA.** The process of a generic MOEA involves initialization, parent selection, recombination, improvement and population update (Verma et al., 2021). For MOCOPs, improvement is a key process to enhance the PF quality. This motivates us to devise an learning-based improvement component in place of traditional one to seek high-quality individuals within the population.

## 4 METHODOLOGY

The L2I framework collaboratively performs efficient local improvements and replaces the traditional problem-specific heuristics. It contains a policy network that generates a node pair of local operations to potentially improving a batch of individual solutions. In what follows, we take MOTSP as an example to elaborate the details of L2I. It is not difficult to generalize to other MOCOPs.

### 4.1 DRL FORMALUTION

L2I learns an improvement policy with respect to a given weight vector so as to approximate the entire PF. The improvement process can be deemed as a Markov decision process (MDP) as follows.

**State.** For an MOTSP instance with $N$ nodes at iteration $t$, the state $s_t$ includes the features of instance $v$, current solution, represented as a sequence $x_t$ with length $N$, and weight vector $\lambda$, i.e., $s_t = \{v^1, \cdots, v^N, x_t^1, \cdots, x_t^N, \lambda\}$, where $v^i$ is the coordinate of node $i$ and $x_t^i$ is the $i$-th visited node in sequence $x_t$.

**Action.** The action $a_t = (i, j)$ is denoted as the node pair $(i, j)$ of a pair-wise local search operator to be conducted. In MOTSP, we adopt an *ensemble* operator, i.e., the combination of three classic operators *relocate*, *exchange*, *2-opt* (see Appendix C for details).

**Transition.** The next state $s_{t+1}$ is obtained by performing action $a_t = (i, j)$ on $x_t$, i.e., selecting the best solution from *relocate*$(i, j)$, *exchange*$(i, j)$ and *2-opt*$(i, j)$.

**Reward.** The reward function is defined by $r_t = f(x_t^*|\lambda) - \min\{f(x_{t+1}|\lambda), f(x_t^*|\lambda)\}$, where $x_t^*$ is the best solution found till iteration $t$ and $r_t > 0$ if an improved solution is found at iteration $t$.

### 4.2 WEIGHT-RELATED POLICY NETWORK

The design of WRPN is based on the encoder-decoder architecture, as illustrated in Figure 1. More details about the network architecture is presented in Appendix B. The policy network first encodes

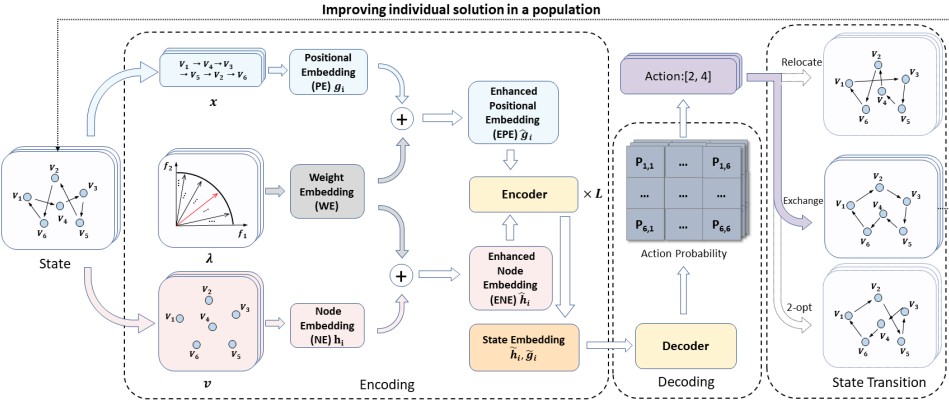

Figure 1: The weight-related policy network (WRPN).

the state into a hidden embedding, and then feeds it into the decoder to compute the action probability matrix. The sampling and greedy decoding strategy are adopted in training and inference, respectively.

The raw features of each solution in the solution population are mapped into two sets of embeddings, including node embeddings and positional embeddings, while the associated weight vector is mapped into weight embeddings.

The node embedding (NE) $h_i$ with dimension $d_h$ is obtained by the linear projection of its node features. The positional embedding (PE) $g_i$ with dimension $d_g$ is initialized by the cyclic positional encoding (Ma et al., 2021) as follows.

$$g_i^{(d)} = \begin{cases} sin(\omega_d \cdot (z(i) \bmod \frac{4\pi}{\omega_d}) - \frac{2\pi}{\omega_d}), \text{if } d \text{ is even} \\ cos(\omega_d \cdot (z(i) \bmod \frac{4\pi}{\omega_d}) - \frac{2\pi}{\omega_d}), \text{if } d \text{ is odd} \end{cases} \tag{1}$$

$$z(i) = \frac{i-1}{N} \frac{2\pi}{\omega_d} \left\lceil \frac{N+1}{2\pi/\omega_d} \right\rceil \tag{2}$$

$$\omega_d = \begin{cases} \frac{3\lfloor d/3 \rfloor}{d_g}(N - N^{\frac{1}{\lfloor d_g/2 \rfloor}}) + N^{\frac{1}{\lfloor d_g/2 \rfloor}}, & \text{if } d < \lfloor d_g/2 \rfloor \\ N, & \text{otherwise} \end{cases} \tag{3}$$

Here, $g_i^{(d)}$ ($i = 1, \ldots, N, d = 1, \ldots, d_g$) is the $d$-th dimension of $g_i$, $z(i)$ is a pattern to make $N$ nodes linearly spaced, and $\omega_d$ is the angular frequency. The weight embedding (WE) with dimension $d_w$ is obtained by the corresponding weight features through a linear projection.

Since a weight vector is applied to guide the search direction, the enhanced node embedding (ENE) $\hat{h}$ is obtained by fusing WE and NE through a feature-wise linear modulation (FiLM) (Brockschmidt, 2020). Similarly, the enhanced positional embedding (EPE) $\hat{g}$ is obtained by the fusion of WE and PE. Such design can better represent the node features and solution features under a weight vector.

### 4.2.1 ENCODER

There are $L$ ($L = 3$) Transformer-style stacked encoders. For better readability, we omit the superscript $l$ for the $l$-th stacked encoder. Instead, we use $\hat{h}$ and $\hat{g}$ to indicate the $l$-th input embeddings, and use $\tilde{h}$ and $\tilde{g}$ to indicate the $l$-th output embeddings, which are also equivalent to the $(l+1)$-th input embeddings.

In each stacked encoder, $\hat{h}$ and $\hat{g}$ are fed into a Dual-Aspect Collaborative Attention (DAC-Att) layer (Ma et al., 2021), followed by the batch normalization, and two separate feed-forward network (FFN) sub-layers, respectively. Because $\hat{h}$ and $\hat{g}$ are originated from two sources, DAC-Att model exhibits good performance compared with the vanilla attention model (Kool et al., 2019).

**DAC-Att layer.** Based on the trainable matrices $W_m^{Q_g} \in \mathbb{R}^{d_{\hat{g}} \times d_q}$, $W_m^{K_g} \in \mathbb{R}^{d_{\hat{g}} \times d_q}$, $W_m^{Q_h} \in \mathbb{R}^{d_{\hat{h}} \times d_k}$ and $W_m^{K_h} \in \mathbb{R}^{d_{\hat{h}} \times d_k}$ ($d_q = d_k = d_{\hat{h}}/m = d_{\hat{g}}/m$, $m = 4$ is the number of attention heads), the

DAC-Att layer computes the node attention score $\alpha_{i,j,m}^h$ and the positional attention score $\alpha_{i,j,m}^g$ for each head $m$, which are given by Eq. (4).

$$\alpha_{i,j,m}^h = \frac{1}{\sqrt{d_k}}(\hat{h}_i W_m^{Q_h})(\hat{h}_j W_m^{K_h})^T, \quad \alpha_{i,j,m}^g = \frac{1}{\sqrt{d_k}}(\hat{g}_i W_m^{Q_g})(\hat{g}_j W_m^{K_g})^T, \tag{4}$$

The $\alpha_{i,j,m}^h$ and $\alpha_{i,j,m}^g$ of each head are concatenated into $\alpha_{i,j}^h$ and $\alpha_{i,j}^g$, then further normalized to $\tilde{\alpha}_{i,j}^h$ and $\tilde{\alpha}_{i,j}^g$ via Softmax. Finally, the output embedding $\tilde{h}_i$ and $\tilde{g}_i$ are computed by Eq. (5–6) with trainable matrix $W^{V_h}, W^{V_g}, W^{R_h}, W^{R_g} \in \mathbb{R}^{d_{\tilde{h}} \times d_v}$ and $W^{O_h}, W^{O_g} \in \mathbb{R}^{2d_{\tilde{h}} \times d_v}$ ($d_v = d_{\tilde{h}}$).

$$\tilde{h}_i = concat[\sum_{j=1}^N \tilde{\alpha}_{i,j}^h(\hat{h}_j W^{V_h}), \sum_{j=1}^N \tilde{\alpha}_{i,j}^g(\hat{h}_j W^{R_h})]W^{O_h} \tag{5}$$

$$\tilde{g}_i = concat[\sum_{j=1}^N \tilde{\alpha}_{i,j}^g(\hat{g}_j W^{V_g}), \sum_{j=1}^N \tilde{\alpha}_{i,j}^h(\hat{g}_j W^{R_g})]W^{O_g} \tag{6}$$

**FFN.** It consists of two linear layers and one 64-dimensional hidden sub-layer with the ReLU activation function.

### 4.2.2 DECODER

In the decoder, the max-pooling and multi-head attention (MHA) modules are applied to independently generate the node-pair selection proposals for $\tilde{h}$ and $\tilde{g}$, after which the respective outputs are aggregated by a multi-layer perception (MLP).

**Max-pooling.** Two independent max-pooling sub-layers are adopted to aggregate the global feature representation for $\tilde{h}$ and $\tilde{g}$, respectively.

**MHA.** The MHA layer effectively represents the attention correlations for each node pair. Given the state embeddings $(\tilde{h}, \tilde{g})$, the correlations score matrices $Y^h, Y^g \in \mathbb{R}^{N \times N}$ are computed as the dot product of the query and key matrices similarly to Kool et al. (2019). Thereafter, the infeasible node pairs are masked as $-\infty$ before Softmax.

**MLP.** We adopt a four-layer MLP with structure $(2m \times 32 \times 32 \times 1)$ to aggregate two node-pair selection proposals from $\tilde{h}$ and $\tilde{g}$.

### 4.3 A SHARED BASELINE FOR PROXIMAL POLICY OPTIMIZATION

We use $n$-step proximal policy optimization (PPO) (Savelsbergh, 1990) to train the policy network and further design a shared baseline to reduce the variance of the training. The overall training process is presented in Algorithm 1.

The training process involves a total of $E$ epochs and $B$ batches per epoch. For each batch, a set of training instances $\mathcal{D}$ is randomly generated (line 3) and a population of weight vectors is randomly sampled from the uniform distribution (line 4). The *curriculum learning* (CL) strategy (Bengio et al., 2009) is used to derive the initial state for better sample efficiency (lines 6–8). The $n$-step return estimation is then exploited to achieve a trade-off between effective reward propagation and bias-variance following the original design of $n$-step PPO (lines 14–18).

We design a shared baseline $b_{t'}^{share}$ (lines 16–17), rather than directly use a baseline obtained by greedy rollout or an extra critic network. The shared baseline and reinforcement learning loss are given in Eq. (7) and Eq. (8), respectively.

$$b_{t'}^{share} = \frac{1}{P}\sum_{p=1}^P R_{t'}^p \tag{7}$$

$$\bigtriangledown_\theta \mathcal{J}(\theta) = \frac{1}{n|\mathcal{D}|}\sum_{\mathcal{D}}\sum_{t'=t}^{t+n} \min\{\frac{\pi_\theta(a_{t'}|s_{t'})}{\pi_{old}(a_{t'}|s_{t'})}A_{t'}, \\ clip[\frac{\pi_\theta(a_{t'}|s_{t'})}{\pi_{old}(a_{t'}|s_{t'})}, 1-\varepsilon, 1+\varepsilon]A_{t'}\} \tag{8}$$

where $P$ is population size, $R_{t'}^p$ is the cumulative reward of offspring $p$ after iteration $t'$ steps and $A_{t'}$ is the advantage compared to $b_{t'}^{share}$. Such a shared baseline exploiting all solutions of the population can induce less variance due to the zero-mean advantage, and also can be computed more efficiently.

---

**Algorithm 1** The training process of WRPN.

---

**Input:** Initial policy network $\pi_\theta$, clipping threshold $\varepsilon$, population size $P$, weight vector distribution $\Lambda$; learning rate $\eta_\theta$, learning rate decay $\beta$, mini-batch $\kappa$, training steps $T_{train}$, CL scalar $\rho^{CL}$.
**Output:** Trained policy network $\pi_\theta$.

1:  **for** $e \leftarrow 1 : E$ **do**
2:    **for** $b \leftarrow 1 : B$ **do**
3:        Randomly generate training instances $\mathcal{D}$;
4:        $\lambda_p \leftarrow \text{SampleWeight}(\Lambda), \forall p \in \{1, \cdots, P\}$;
5:        $\delta_p \leftarrow \text{InitialSolutions}(\mathcal{D}), \forall p \in \{1, \cdots, P\}$;
6:        Improve $\delta_p$ via $\{(\pi_\theta, \lambda_p)\}$ for $T = e/\rho^{CL}$ steps;
7:        $s_0 \leftarrow \text{InitialState}(\delta_p, \lambda_p), \forall p \in \{1, \cdots, P\}; t \leftarrow 0$;
8:        **while** $t < T_{train}$ **do**
9:            Get $\{(s_{t'}, a_{t'}, r_{t'})\}_{t'=t}^{t+n}$ where $a_{t'} \sim \pi_\theta(a_{t'}|s_{t'})$;
10:           $t \leftarrow t + n, \pi_{old} \leftarrow \pi_\theta$;
11:           **for** $z \leftarrow 1 : \kappa$ **do**
12:               $R_{t+1} = 0$;
13:               **for** $t' \in \{t, t-1, \cdots, t-n\}$ **do**
14:                   $R_{t'} \leftarrow r_t(\lambda) + \gamma R_{t'+1}$;
15:                   Compute baseline $b_{t'}^{share}$ using Eq. (7);
16:                   $A_{t'} \leftarrow R_{t'} - b_{t'}^{share}$;
17:               **end for**
18:               Compute RL loss $\mathcal{J}(\theta)$ using Eq. (8);
19:               $\theta \leftarrow \theta + \eta_\theta \bigtriangledown_\theta \mathcal{J}(\theta)$;
20:           **end for**
21:        **end while**
22:     **end for**
23:     $\eta_\theta \leftarrow \beta\eta_\theta$
24: **end for**

**Note:** The superscipt $p$ of $s^p, a^p, A^p, r^p$ and $R^p$ for individual $p$ is omitted from lines 7 to 16 for readability.

---

### 4.4 QUALITY ENHANCEMENT

To further enhance the proximity and diversity of the Pareto set, we propose a quality enhancement mechanism based on *instance augmentation* (Lin et al., 2022). Specifically, an instance of MOCOPs still retains the equivalent optimal solution after multiple transformations, such as spatial rotations and reflections. These instances are then solved by an MOEA with the trained WRPN to further promote exploration. The final external population is obtained by aggregating all the non-dominated solutions during search with respect to different transformed instances. More details of the algorithm implementation can be found in Appendix G.

## 5 EXPERIMENTS

The experiments are conducted on a server with an Intel E5-2678 v3 CPU @ 2.50GHz and 8 TITAN Xp GPUs. All the compared methods are implemented in Python using the Pytorch library, except that **LKH** (Helsgaun, 2000; Tinós et al., 2018) is in C-style. Our codes will be made publicly available.

### 5.1 EXPERIMENTAL SETUP

**Problems.** We introduce two classic MOCOPs, i.e., multiobjective traveling salesman problem (MOTSP) (Lust and Teghem, 2010) and multiobjective capacitated vehicle routing problem (MOCVRP) (Jozefowiez et al., 2008). Concretely, we consider bi/tri-objective (Bi/Tri-TSP) and bi-objective CVRP (Bi-CVRP). For $M$-objective MOTSP, $M$ groups of coordinates are given to define $M$ Euclidean travelling costs between a pair of nodes. The objective is to simultaneously minimize $M$ total costs. For Bi-CVRP, the objective is to minimize the total route lengths, as well as the length of the longest route. See Appendix A for detailed descriptions of MOTSP and MOCVRP.

**Compared approaches.** We compare our WRPN with three kinds of representative approaches. (1) MOEA approaches: **MOEA/D** and **NSGA-II** with 5000 generations and **MOGLS** with 2000 generations and 50 local improvements; **PPLS/D-C** (Shi et al., 2022) with 200 generations. All of them are efficiently implemented in parallel. These MOEAs also use *relocate*, *exchange*, and *2-opt* operator for MOTSP and MOCVRP. The population sizes of the former three are set identically to 100, while the subregion number of PPLS/D-C is set to 10 according to the original paper. (2) L2C approaches: **DRL-MOA** (Li et al., 2020), **POMO-T** (Kwon et al., 2020), **PMOCO** (Lin et al., 2022) and **MLDRL** (Zhang et al., 2022). The former two use decomposition and parameter migration strategies with Point Network (Vinyals et al., 2015) and POMO (Kwon et al., 2020) as their single-objective submodels, respectively. The third one is the preference-conditioned multiobjective combinatorial optimization in which the submodel is also POMO. The last one is the meta-learning approach with POMO as its model architecture. Note that all the above approaches consider 101 decomposed subproblems. (3) The state-of-the-art single-objective solvers under WS scalarization. 101 decomposed subproblems are considered. Each subproblem is solved by **LKH** or **OR-Tools**[1] for MOTSP.

**Metrics.** We primarily adopt the hypervolume (HV) (Zitzler et al., 2007), the number of non-dominated solutions ($|$NDS$|$) and the inverted generational distance plus (IGD$^+$) (Ishibuchi et al., 2016) to evaluate the performance of the compared algorithms. In general, the larger the HV is and the smaller the IGD$^+$ is, the better the corresponding algorithm performs. Our proposed method is highlighted in *italic* and the best result and its statistically indifferent results are highlighted in **bold**. The second-best result and the one without statistical significance to it are highlighted as underline. A Wilcoxon rank-sum test with a significance level 1% is applied to compare the experimental results. Additional details of the above metrics are introduced in Appendix D.

**Training details.** For MOTSP with different scales, the model is trained with $E = 200$ epochs. Each epoch has $B = 90$ batches with batch size 120. For MOCVRP, we respectively set $E = 200/200/100$ with batch size $120/100/32$ for MOCVRP-20/50/100. The population size $P$ is set to 20 in training. We set PPO-step $n = 4$ and $T_{train} = 200$ for MOTSP, while set $n = 5$ and $T_{train} = 500$ for MOCVRP. The gradient norm is clipped within $\varepsilon = 0.04/0.2/0.45$ for the problems with scale $20/50/100$. The reward discount factor is set to $\gamma = 0.999$. The Adam optimizer with a learning rate $\eta_\theta = 10^{-4}$ is adopted, while decaying with $\beta = 0.985$ per epoch. To accelerate convergence, the pre-trained 50-size model is used to train 100-size model. We also implement GPU parallel training. For MOTSP-20/50/100, an epoch roughly takes $7/22/40$ minutes, respectively. For MOCVRP-20/50/100, it takes about $17/30/50$ minutes, respectively.

**Inference details.** During inference, the population size $P$ is set to 100. The number of iterations $T$ is set to $2000/5000$. The number of local operations per iteration is set to 50, except for MOCVRP-100 that is increased to 100 due to its difficulty. All the compared methods are evaluated on the same test dataset with 200 random instances. We further test the performance of PMOCO and MLDRL with instance augmentation, in which 8 transformed instances are considered. Similarly, our WRPN module based on MOGLS is also equipped with quality enhancement using 8 transformed instances.

## 5.2 RESULTS

**MOTSP.** For each problem size, the average IGD$^+$, average HV, gap to the best HV, number of non-dominated solutions, and average (inference) time on 200 random instances are shown in Table 1. Compared with basic MOEA approaches, WRPN incorporated with different MOEAs have outstanding performance. In particular, MOGLS+WRPN achieves the lowest gap in fewest time for all problem sizes. The comprehensive metrics, HV and IGD$^+$, both indicate that WRPN is excellent in proximity and diversity. When quality enhancement mechanism is adopted, MOGLS+WRPN(AUG) with 5K iterations attains the highest HV for all sizes. It is even better than WS-LKH which is based on weight decomposition. Moreover, our WRPN achieves larger gaps compared with other competitors on Tri-TSP, e.g., a 2.94% gap for WS-LKH on Tri-TSP-100, which shows that WRPN is able to excavate more potential Pareto solutions in the higher-dimensional objective space.

It is worth noting that WRPN may generate more than $100/105$ non-dominated solutions. Its HV can take advantage of the mass number of solutions. In Appendix F.2, we further analyze the effect of the number of non-dominated solutions on HV.

---

[1] https://developers.google.com/optimization/

Table 1: Comparison results on 200 random MOTSP instances.

| Method | Bi-TSP-20 | | | | | Bi-TSP-50 | | | | | Bi-TSP-100 | | | | |
|---|---|---|---|---|---|---|---|---|---|---|---|---|---|---|---|
| | IGD$^+$ | HV | GAP | \|NDS\| | TIME | IGD$^+$ | HV | GAP | \|NDS\| | TIME | IGD$^+$ | HV | GAP | \|NDS\| | TIME |
| WS-LKH | 0.23 | 0.6270 | 0.44% | 15 | 3s | 0.19 | 0.6415 | 0.56% | 38 | 30s | 0.12 | **0.7090** | **0.00%** | 65 | 1.8m |
| WS-ORTools | 0.22 | 0.6256 | 0.67% | 20 | 2s | 0.30 | 0.6353 | 1.52% | 49 | 16s | 0.40 | 0.7004 | 1.21% | 66 | 78s |
| MOEA/D(T=5K) | 0.35 | 0.6264 | 0.81% | 16 | 47s | 0.56 | 0.6336 | 1.78% | 56 | 72s | 0.71 | 0.6951 | 1.96% | 181 | 2.2m |
| NSGA-II(T=5K) | 0.17 | 0.6283 | 0.24% | 77 | 2.3m | 1.07 | 0.6145 | 4.74% | 98 | 2.3m | 2.09 | 0.6634 | 6.43% | 42 | 2.6m |
| MOGLS(T=5K) | 0.18 | 0.6287 | 0.17% | 49 | 14s | 0.48 | 0.6296 | 2.40% | 88 | 18s | 2.30 | 0.6506 | 8.24% | 103 | 31s |
| PPLS/D-C(T=200) | 0.18 | 0.6256 | 0.67% | 71 | 68s | 0.45 | 0.6282 | 2.62% | 213 | 6.4m | 1.07 | 0.6844 | 3.47% | 372 | 31.2m |
| DRL-MOA | 0.78 | 0.5973 | 5.16% | 17 | 2s | 1.50 | 0.5909 | 8.40% | 31 | 4s | 3.40 | 0.6390 | 9.87% | 38 | 7s |
| POMO-T | 0.24 | 0.6257 | 0.65% | 23 | 2s | 0.33 | 0.6360 | 1.41% | 57 | 4s | 0.60 | 0.6970 | 1.69% | 70 | 7s |
| PMOCO | 0.20 | 0.6259 | 0.62% | 18 | 2s | 0.31 | 0.6351 | 1.55% | 49 | 4s | 0.54 | 0.6957 | 1.88% | 71 | 7s |
| MLDRL | 0.20 | 0.6271 | 0.43% | 26 | 2s | 0.28 | 0.6364 | 1.35% | 63 | 4s | 0.56 | 0.6969 | 1.71% | 73 | 7s |
| *MOEA/D+WRPN(T=5K)* | **0.14** | 0.6297 | 0.02% | 76 | 95s | 0.09 | 0.6445 | 0.09% | 326 | 2.5m | 0.20 | 0.7061 | 0.41% | 258 | 4.6m |
| *NSGA-II+WRPN(T=5K)* | **0.14** | 0.6296 | 0.03% | 93 | 2.8m | 0.11 | 0.6433 | 0.28% | 272 | 4.4m | 0.22 | 0.7057 | 0.47% | 234 | 5.8m |
| *MOGLS+WRPN(T=2K)* | **0.14** | 0.6296 | 0.03% | 76 | 15s | 0.09 | 0.6443 | 0.12% | 241 | 22s | 0.22 | 0.7055 | 0.49% | 192 | 44s |
| *MOGLS+WRPN(T=5K)* | **0.14** | 0.6297 | 0.02% | 84 | 37s | 0.08 | 0.6446 | 0.08% | 302 | 55s | 0.16 | 0.7069 | 0.30% | 233 | 1.8m |
| PMOCO(AUG) | 0.23 | 0.6271 | 0.43% | 15 | 2s | 0.16 | 0.6401 | 0.78% | 51 | 5s | 0.33 | 0.7013 | 1.09% | 77 | 12s |
| MLDRL(AUG) | 0.23 | 0.6271 | 0.43% | 16 | 2s | 0.13 | 0.6408 | 0.67% | 63 | 5s | 0.34 | 0.7022 | 0.96% | 83 | 12s |
| *MOGLS+WRPN(AUG)(T=2K)* | **0.14** | **0.6298** | **0.00%** | 92 | 27s | 0.01 | 0.6450 | 0.02% | 490 | 60s | 0.10 | 0.7081 | 0.13% | 324 | 2.7m |
| *MOGLS+WRPN(AUG)(T=5K)* | **0.14** | **0.6298** | **0.00%** | 94 | 66s | **0.00** | **0.6451** | **0.00%** | 465 | 2.7m | **0.05** | **0.7090** | **0.00%** | 393 | 6.9m |

| Method | Tri-TSP-20 | | | | | Tri-TSP-50 | | | | | Tri-TSP-100 | | | | |
|---|---|---|---|---|---|---|---|---|---|---|---|---|---|---|---|
| | IGD$^+$ | HV | GAP | \|NDS\| | TIME | IGD$^+$ | HV | GAP | \|NDS\| | TIME | IGD$^+$ | HV | GAP | \|NDS\| | TIME |
| WS-LKH | 0.24 | 0.4712 | 1.34% | 61 | 4s | 0.51 | 0.4440 | 3.42% | 103 | 33s | 0.82 | 0.5076 | 2.94% | 105 | 2m |
| WS-ORTools | 0.23 | 0.4703 | 1.53% | 76 | 3s | 0.71 | 0.4348 | 5.42% | 102 | 19s | 1.26 | 0.4947 | 5.41% | 105 | 83s |
| MOEA/D(T=5K) | 0.23 | 0.4704 | 1.51% | 104 | 48s | 0.61 | 0.4376 | 4.81% | 627 | 78s | 2.19 | 0.4698 | 10.17% | 978 | 2.3m |
| NSGA-II(T=5K) | 0.37 | 0.4572 | 4.27% | 527 | 2.2m | 3.55 | 0.3295 | 28.32% | 624 | 3.1m | 8.48 | 0.3301 | 36.88% | 588 | 3.4m |
| MOGLS(T=2K) | 0.16 | 0.4722 | 1.13% | 242 | 15s | 1.49 | 0.3958 | 13.90% | 448 | 21s | 5.52 | 0.3789 | 27.55% | 496 | 32s |
| PPLS/D-C(T=200) | 0.11 | 0.4698 | 1.63% | 875 | 3.4m | 0.96 | 0.4174 | 9.20% | 3723 | 19.3m | 3.02 | 0.4376 | 16.33% | 8096 | 1.1h |
| PMOCO | 0.25 | 0.4693 | 1.74% | 71 | 2s | 0.77 | 0.4315 | 6.14% | 103 | 4s | 1.57 | 0.4858 | 7.11% | 105 | 7s |
| MLDRL | 0.24 | 0.4701 | 1.61% | 72 | 2s | 0.74 | 0.4317 | 6.10% | 103 | 4s | 1.57 | 0.4852 | 7.23% | 104 | 7s |
| *MOEA/D+WRPN(T=5K)* | 0.08 | 0.4734 | 0.88% | 347 | 95s | 0.35 | 0.4485 | 2.44% | 683 | 2.6m | 0.67 | 0.5091 | 2.66% | 709 | 4.3m |
| *NSGA-II+WRPN(T=5K)* | 0.13 | 0.4734 | 0.88% | 543 | 3.8m | 0.35 | 0.4474 | 2.68% | 5033 | 4.4m | 1.44 | 0.4948 | 5.39% | 7074 | 7.6m |
| *MOGLS+WRPN(T=2K)* | 0.08 | 0.4759 | 0.36% | 486 | 15s | 0.24 | 0.4528 | 1.50% | 1269 | 26s | 0.55 | 0.5115 | 2.20% | 1436 | 49s |
| *MOGLS+WRPN(T=5K)* | 0.06 | 0.4766 | 0.21% | 693 | 40s | 0.17 | 0.4553 | 0.96% | 2190 | 59s | 0.38 | 0.5159 | 1.36% | 2478 | 2.2m |
| PMOCO(AUG) | 0.24 | 0.4712 | 1.34% | 62 | 7s | 0.57 | 0.4409 | 4.09% | 104 | 19s | 1.19 | 0.4956 | 5.24% | 105 | 86s |
| MLDRL(AUG) | 0.24 | 0.4712 | 1.34% | 61 | 7s | 0.55 | 0.4408 | 4.11% | 104 | 19s | 1.16 | 0.4958 | 5.20% | 105 | 86s |
| *MOGLS+WRPN(T=2K)(AUG)* | 0.03 | 0.4774 | 0.04% | 1078 | 38s | 0.06 | 0.4587 | 0.23% | 4629 | 90s | 0.16 | 0.5206 | 0.46% | 5327 | 3.8m |
| *MOGLS+WRPN(T=5K)(AUG)* | **0.01** | **0.4776** | **0.00%** | 1358 | 91s | **0.03** | **0.4597** | **0.00%** | 4243 | 3.6m | **0.06** | **0.5230** | **0.00%** | 8007 | 9.2m |

Table 2: Comparison results on 200 random MOCVRP instances.

| Method | Bi-CVRP-20 | | | | | Bi-CVRP-50 | | | | | Bi-CVRP-100 | | | | |
|---|---|---|---|---|---|---|---|---|---|---|---|---|---|---|---|
| | IGD$^+$ | HV | GAP | \|NDS\| | TIME | IGD$^+$ | HV | GAP | \|NDS\| | TIME | IGD$^+$ | HV | GAP | \|NDS\| | TIME |
| MOEA/D(T=5K) | 0.68 | 0.3913 | 9.19% | 8 | 7m | 0.33 | 0.4012 | 2.38% | 6 | 22.2m | 0.88 | 0.3976 | 2.07% | 7 | 38.5m |
| NSGA-II(T=5K) | 0.06 | 0.4273 | 0.84% | 14 | 5.7m | 0.75 | 0.3937 | 4.23% | 14 | 10.2m | 1.67 | 0.3821 | 5.89% | 13 | 14.7m |
| MOGLS(T=5K) | 0.04 | 0.4285 | 0.56% | 11 | 96s | 0.44 | 0.3992 | 2.87% | 11 | 4.2m | 1.14 | 0.3939 | 2.98% | 11 | 6.6m |
| PPLS/D-C(T=200) | 0.04 | 0.4287 | 0.51% | 15 | 24.6m | 0.31 | 0.4007 | 2.51% | 17 | 4.1h | 1.00 | 0.3946 | 2.81% | 20 | 14.8h |
| POMO-T | 0.03 | 0.4287 | 0.51% | 7 | 3s | 0.04 | 0.4076 | 0.83% | 10 | 7s | 0.16 | 0.4055 | 0.12% | 12 | 14s |
| PMOCO | 0.06 | 0.4267 | 0.97% | 5 | 3s | 0.08 | 0.4031 | 1.92% | 6 | 6s | 0.24 | 0.3908 | 3.74% | 5 | 13s |
| MLDRL | 0.04 | 0.4181 | 2.97% | 5 | 3s | 0.04 | 0.4020 | 2.19% | 9 | 6s | 0.17 | 0.4022 | 0.94% | 11 | 13s |
| *MOEA/D+WRPN(T=5K)* | 0.01 | 0.4304 | 0.09% | 16 | 8.2m | 0.03 | 0.4092 | 0.23% | 20 | 32.1m | 0.21 | 0.4022 | 0.54% | 16 | 40.8m |
| *NSGA-II+WRPN(T=5K)* | 0.01 | 0.4295 | 0.32% | 15 | 6.4m | 0.08 | 0.4075 | 0.85% | 19 | 12m | 0.29 | 0.4045 | 0.37% | 26 | 20.2m |
| *MOGLS+WRPN(T=2K)* | 0.01 | 0.4307 | 0.05% | 14 | 53s | 0.03 | 0.4097 | 0.32% | 18 | 2.2m | 0.34 | 0.4034 | 0.64% | 18 | 6.9m |
| *MOGLS+WRPN(T=5K)* | 0.01 | 0.4308 | 0.02% | 15 | 2.2m | 0.02 | 0.4101 | 0.22% | 20 | 4.6m | 0.30 | 0.4044 | 0.39% | 20 | 17.4m |
| PMOCO(AUG) | 0.04 | 0.4294 | 0.35% | 6 | 3s | 0.04 | 0.4077 | 0.80% | 7 | 7s | 0.18 | 0.3966 | 2.32% | 7 | 14s |
| MLDRL(AUG) | 0.03 | 0.4219 | 2.09% | 6 | 3s | 0.02 | 0.4065 | 1.09% | 10 | 7s | **0.14** | 0.4059 | 0.02% | 13 | 14s |
| *MOGLS+WRPN(AUG)(T=2K)* | **0.00** | **0.4309** | **0.00%** | 17 | 84s | **0.01** | 0.4107 | 0.07% | 24 | 4.1m | 0.28 | 0.4052 | 0.20% | 24 | 16.5m |
| *MOGLS+WRPN(AUG)(T=5K)* | **0.00** | **0.4309** | **0.00%** | 18 | 3.5m | **0.01** | **0.4110** | **0.00%** | 26 | 10m | 0.24 | **0.4060** | **0.00%** | 27 | 41m |

**MOCVRP.** Table 2 presents the results for MOCVRP instances. From the table, we can find that MOEAs equipped with WRPN are significantly better than MOEAs in terms of both running time and solution quality. It also outperforms advanced L2C methods in terms of solution quality. Although WRPN generally requires longer time, its training efficiency is better than L2C methods. We have verified that the number of training samples of WRPN is about one to two orders of magnitude less than PMOCO and MLDRL. Finally, it is worth noting that MOCVRP is more difficult to solve than

MOTSP. In many leading heuristic approaches, complex problem-specific operators are proposed to locally improve a solution. In our WRPN methods for MOCVRP, the *ensemble* operator only includes *relocate*, *exchange* and *2-opt* (see Appendix C for details). Such a simple design can partly validate the effectiveness of WRPN.

### 5.3 ABLATION STUDY

**Effects of the weight-related policy network.** To verify our design of the weight-related policy network, we compare WRPN with DACT (Ma et al., 2021), a representative L2I method for solving single-objective COPs. DACT cannot directly address MOCOPs, since its policy network cannot handle the weight vector. Therefore, we combine DACT with transfer learning (Li et al., 2020) to cope with decomposed subproblems, thus solving MOCOPs. Specifically, we adopt the DACT model pre-trained by the authors and perform transfer learning with 5 epochs per subproblem (keeping the training dataset size consistent with WRPN). The final results based on the MOGLS framework are displayed in Table 3. It can be observed that DACT exhibits poor performance with expensive inference time. We also compare our WRPN with DACT on single objective that DACT specializes in. The results can be found in Appendix H.

**Effects of the shared baseline.** To highlight the effectiveness of our shared baseline in training, we compare it with the critic network as baseline (Ma et al., 2021) on 200 MOTSP-50 instances, while keeping other settings of the model unchanged. For the critic baseline, it consists of one MHA layer, two pooling layers and one three-layer MLP, which produces an estimated value for the current state by using the encoder output $(\tilde{h}, \tilde{g})$. Figure 2 shows that the model trained with our shared baseline converges much faster than that with the critic baseline.

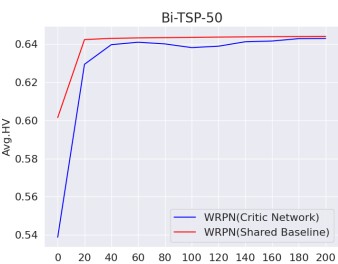

Figure 2: Effects of the shared baseline.

Table 3: Effects of the weight-related policy network.

| Method | Bi-VRP-20 | | |
| --- | --- | --- | --- |
| | HV | GAP | TIME |
| DACT-T(T=2K) | 0.3979 | 7.66% | 1.5h |
| *WRPN(T=2K)* | 0.4307 | 0.05% | 53s |
| *WRPN(AUG)(T=5K)* | **0.4309** | **0.00%** | 84s |

Table 4: Effects of the quality enhancement.

| Method | Bi-TSP-100 | | |
| --- | --- | --- | --- |
| | HV | GAP | TIME |
| WRPN (VIA) (T=2K) | 0.7018 | 1.02% | 2.5m |
| WRPN (VIA) (T=5K) | 0.7044 | 0.65% | 5.7m |
| WRPN (QE) (T=2K) | 0.7081 | 0.13% | 2.7m |
| WRPN (QE) (T=5K) | **0.7090** | **0.00%** | 6.9m |

**Effects of the quality enhancement.** To provide further insights and comparisons, we conducted complementary experiments to evaluate the impact of the proposed quality enhancement (QE) technique and a vanilla instance augmentation (VIA) technique proposed by Lin et al. (2022). The VIA technique simply retains the best solution of multiple transformed instances in a prescribed direction during each iteration, without utilizing a set of temporal external populations to archive non-dominated solutions. The results based on MOGLS are presented in Table 4. It shows that our QE mechanism outperforms the VIA technique in terms of HV. This finding clearly demonstrates the utility of QE as an effective mechanism for enhancing diversity exploration in MOCOPs. More experimental results can be found in Appendix G.

## 6 CONCLUSION

L2I is a generic DRL-based improvement paradigm for MOCOPs that iteratively improve a population of solutions. We propose WRPN, an end-to-end model, to effectively guide the local improvement. A shared baseline is designed to train WRPN efficiently. The quality enhancement mechanism is also adopted to improve the search. Extensive experiments on MOTSP and MOCVRP justify that WRPN can produce promising PF with good proximity and diversity, and achieve state-of-the-art results. Note that WRPN is also applicable to other MOEA frameworks. In future, more advanced DRL techniques can be devised to better learn implicit patterns of MOEA components. In addition, it would be interesting to investigate the performance of other L2I approaches to solve complex MOCOPs with many objectives.

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

# A PROBLEM DESCRIPTIONS

## A.1 MOTSP

**Definition.** The multiobjective traveling salesman problem (MOTSP) is one of the most commonly studied MOCOPs. For MOTSP with $N$ nodes and $M$ objectives, the $i$-th node has $M$ 2-dimensional coordinates $\{(x_{i,1}^1, x_{i,2}^1), \cdots, (x_{i,1}^M, x_{i,2}^M)\}$. The $m$-th cost $c_{ij}^m = \left\| x_i^m - x_j^m \right\|_2$ is the Euclidean distance between node $i$ and node $j$. The goal is to find a cyclic permutation $\pi$ to minimize all the $M$ sum costs simultaneously.

$$\min \ f(\pi) = (f_1(\pi), f_2(\pi), \cdots, f_M(\pi)), \tag{9}$$

where

$$f_m(\pi) = c_{\pi_N, \pi_1}^m + \sum_{i=1}^{N-1} c_{\pi_i, \pi_{i+1}}^m. \tag{10}$$

**Training Instances.** For each instance, $N$ nodes are randomly sampled in the $2M$-dimensional unit hyper-square $[0, 1]^{2M}$ with the uniform distribution.

## A.2 MOCVRP

**Definition.** The capacitated vehicle routing problem (CVRP) is a classic extension of TSP. In addition to $N$ customer nodes, CVRP has a depot node. Each node is associated with 2-dimensional Euclidean coordinates. The customer node $i$ has a demand $d_i$ to be satisfied. An unlimited number of delivery vehicles with fixed capacity $D$ starts from the depot, then deliver goods to customer nodes to satisfy their demands, and finally returns to the depot. Each customer must be served exactly once and the total demands served by each vehicle cannot exceed $D$.

In the multiobjective capacitated vehicle routing problem (MOCVRP), we consider two objectives, i.e., the total route length and the length of the longest route (also known as *makespan*).

**Training Instances.** The coordinates of $n$ customer nodes and the depot node are randomly sampled in the unit square with the uniform distribution. The demand $d_i$ of customer $i$ is sampled uniformly from a discret set $\{1, 2, \cdots, 9\}$. The capacity $D$ is set to 30/40/50 for $n = 20/50/100$. Similar to the previous works (Kwon et al., 2020; Lin et al., 2022), the demands and capacity will be normalized to $\hat{d}_i = \frac{d_i}{D}$ and $\hat{D} = \frac{D}{D} = 1$.

# B ADDITIONAL DETAILS OF THE WEIGHT-RELATED POLICY NETWORK

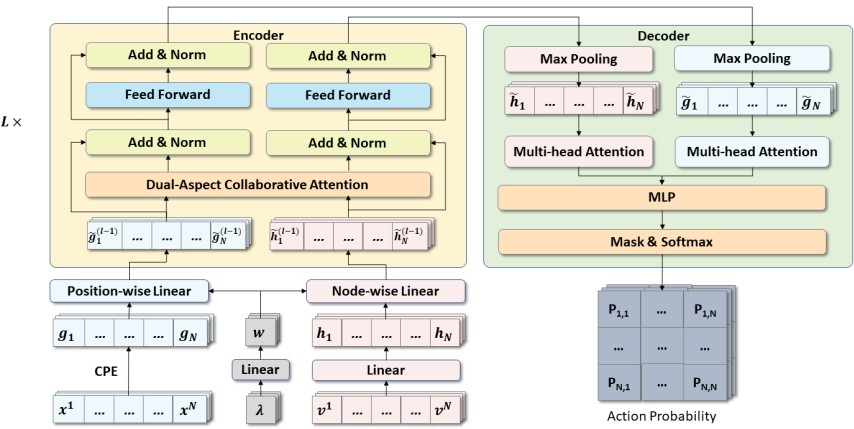

Figure 3: Architecture of the weight-related policy network.

---

**Algorithm 2** Pseudocode of WRPN.

---

**Input:** Solutions set $x$, Instance $v$, Weight vector set $\lambda$, Number of stacked encoders $L$.
**Output:** Improved solutions set $x'$.
 1: **procedure** WRPN($x$, $\lambda$, $v$)
 2:     $h \leftarrow$ NodeEmbedding ($v$);
 3:     $g \leftarrow$ PositionalEmbedding ($x$);
 4:     $\tilde{\lambda} \leftarrow$ WeightEmbedding ($\lambda$);
 5:     $\tilde{h} \leftarrow$ Node-wise Linear ($h$, $\tilde{\lambda}$);
 6:     $\tilde{g} \leftarrow$ Position-wise Linear ($g$, $\tilde{\lambda}$);
 7:     **for** $l \leftarrow 1 : L$ **do**
 8:         $\tilde{h}^{(l)}, \tilde{g}^{(l)} \leftarrow$ Encoder ($\tilde{h}^{(l-1)}, \tilde{g}^{(l-1)}$);
 9:     **end for**
10:     Action Probability Matrix $P \leftarrow$ Decoder($\tilde{h}^{(L)}, \tilde{g}^{(L)}$);
11:     $a \leftarrow$ GreedyAction($P$);
12:     $x' \leftarrow$ EnsembleOperator($x$, $a$);
13:     **return** $x'$;
14: **end procedure**

---

WRPN generates the action proposal for improving a solution $x$ under a given preference $\lambda$. Algorithm 2 provides the pseudo-code for WRPN to solve an MOCOP instance $v$, where the Node-wise Linear and Position-wise Linear follows the design of FiLM (Brockschmidt, 2020). Figure 3 illustrates the detailed architecture of the policy network, which is based on the encoder-decoder architecture. The proposed network is a unified model for general MOCOPs, with problem-specific adjustments for different inputs and masking mechanisms. In the following, we present the network inputs and masking mechanisms for both MOTSP and MOCVRP.

## B.1   NETWORK INPUTS OF MOTSP

The network inputs of MOTSP contain:

- $N$ $2M$-dimensional vectors for the nodes $v_1, \ldots, v_N$, where each node is associated with $M$ groups of coordinates.

- An $N$-dimensional vector for the current solutions $x_1, \ldots, x_N$.

- A $M$-dimensional vector for the decomposed weight.

We simply mask all the diagonal elements of the output action probability matrix, since they have no meanings for the designed operators.

## B.2   NETWORK INPUTS OF MOCVRP

A solution of MOCVRP may contain multiple routes. We use a *giant tour* representation, which concatenate all the routes using the depot node as delimiters, to denote a solution. For example, a solution with three routes $\{(0, 1, 2, 0), (0, 3, 0), (0, 4, 5, 6, 0)\}$ can be represented as $(0, 1, 2, 0, 3, 0, 4, 5, 6, 0)$. For ease of parallel batch training, we add multiple dummy nodes to the end of solutions to align the length of different solutions (Wu et al., 2021). In this paper, the maximum number of dummy depots is set to $Q = 10/20/20$ for $N = 20/50/100$. Thus, the total number of nodes in a solution is $N + Q + 1$.

The network inputs for MOCVRP contain:

- $N$ 3-dimensional vectors for the nodes $v_1, \ldots, v_N$, where each node is associated with two coordinates and a demand.

- A 2-dimensional vector for the depot $v_0$.

- A 2-dimensional vector for the decomposed weight.

- An $(N + Q + 1)$-dimensional vector for the current solutions $x_1, \ldots, x_{N+Q+1}$.

For MOCVRP, we further design a few more solution-specific features for each node. Given a specific solution $(x_1 = 0, \cdots, x_{i-1}, x_i, x_{i+1}, \cdots, x_{N+Q+1} = 0)$, we include an additional 4-dimensional vector for each node $x_i$, which consists of a 2-dimensional vector for the total demand before node $x_i$ and the total demand after node $x_i$ (inclusive) on the corresponding sub-route, and a 2-dimensional vector for the distance from node $x_i$ to its neighbor nodes $x_{i-1}$ and $x_{i+1}$;

The masking mechanism in MOCVRP is similar to that in MOTSP. In addition to masking the diagonal elements, those elements in the action probability matrix which make the solution infeasible (e.g., violating the capacity constraints) are also masked.

## C  DETAILS OF OPERATORS

A solution of MOTSP or MOCVRP can be represented by a sequence of nodes. The following three operators are applied to define the neighborhood of a solution.

- *Relocate*$(i,j)$: Remove the node at location $i$ and relocate it next to location $j$.

- *Exchange*$(i,j)$: Exchange the node at location $i$ with the node at location $j$.

- *2-opt*$(i,j)$: Exchange two edges by reversing a segment between location $i$ and location $j$.

## D  METRIC DETAILS

**Hypervolume (HV).** HV is a widely used hybrid indicator to evaluate the quality of Pareto fronts (PFs). It represents the volume covered by the non-dominated set of solutions with respect to the reference point. HV comprehensively measures the convergence and diversity of PFs. Given an approximated Pareto set $\mathcal{P} \subset \mathbb{R}^M$ and a reference point $\mathbf{r}^*$, we can calculate the hypervolume HV($\mathcal{P}$, $\mathbf{r}^*$) as defined in Eq. (11).

$$\text{HV}(\mathcal{P}, \mathbf{r}^*) = \text{VOL}(S) \tag{11}$$

$$S = \{\mathbf{r} \in \mathbb{R}^M \mid \exists \mathbf{r} \in \mathcal{P} \text{ such that } y \prec \mathbf{r} \prec \mathbf{r}^*\} \tag{12}$$

where VOL($\cdot$) is the Lebesgue measure (Zizler, 1998), and $\mathbf{r}^*$ is dominated by all solutions in $\mathcal{P}$.

As the HV values vary significantly due to different objective scales of different problems, our experimental results report the normalized values $\tilde{H}(PF, \mathbf{r}^*) = \text{HV}(PF, \mathbf{r}^*)/\prod_{i=1}^{M} \mathbf{r}_i^*$, where all the methods share the same $\mathbf{r}^*$. The specific settings of $\mathbf{r}^*$ for all MOCOPs are shown in Table 5.

Table 5: The settings of reference points for different MOCOPs.

| Problem | Size | Reference point $\mathbf{r}^*$ |
|---------|------|--------------------------------|
| Bi-TSP | 20 | (20, 20) |
| | 50 | (35, 35) |
| | 100 | (65, 65) |
| Bi-CVRP | 20 | (30, 4) |
| | 50 | (45, 4) |
| | 100 | (80, 4) |
| Tri-TSP | 20 | (20, 20, 20) |
| | 50 | (35, 35, 35) |
| | 100 | (65, 65, 65) |

**Inverted generational distance plus (IGD$^+$).** IGD$^+$ is a variant of the inverted generational distance (IGD) that is also widely used to evaluate the performance of multiobjective combinatorial optimization algorithms. Compared with IGD, it has the advantage of computation simplicity and weakly Pareto-compliant, so that it can more accurately evaluate PFs. IGD$^+$ measures the distance between the approximated Pareto solutions $\mathcal{P}$ and the Pareto optimal solutions $\mathcal{P}^*$ in PF. The

definitions of IGD$^+$ is as follows:

$$\text{IGD}^+(\mathcal{P}) = (\sum_{i=1}^{|\mathcal{P}^*|} \min_{j=1}^{|\mathcal{P}|} d(\mathbf{z}_i, \mathbf{x}_j))/|\mathcal{P}^*|, \mathbf{z} \in \mathcal{P}^*, \mathbf{x} \in \mathcal{P} \tag{13}$$

$$d(\mathbf{z}, \mathbf{x}) = \sqrt{(\max\{x_1 - z_1, 0\})^2 + \cdots, (\max\{x_M - z_M, 0\})^2} \tag{14}$$

In above, $x_i$ and $z_i$ denote the $i$-th objective value in the approximation solution $x$ and the Pareto optimal solution $z$, respectively.

Since $\mathcal{P}^*$ is difficult to be known beforehand, all the approximated solutions obtained by different algorithms for the same instance are aggregated to represent $\mathcal{P}^*$ in this paper.

## E    ADDITIONAL DETAILS OF EXPERIMENTS

### E.1    RESULTS ON BENCHMARK INSTANCES

In Table 6, we report results of all the methods on four MOTSP benchmark instances, which are constructed by KroA/B/C/D/E in the TSPLIB (Reinelt, 1991). It can be noted that for all the instances, our WRPN significantly outperforms other methods in terms of HV and IGD$^+$. The visualisation in Figure 4 also validates the superiority of WRPN.

Table 6: Computational results on MOTSP benchmark instances.

| Method | KroAB100 | | | | KroBC100 | | | | KroCD100 | | | | KroDE100 | | | |
|---|---|---|---|---|---|---|---|---|---|---|---|---|---|---|---|---|
| | IGD$^+$ | HV | GAP | \|NDS\| | IGD$^+$ | HV | GAP | \|NDS\| | IGD$^+$ | HV | GAP | \|NDS\| | IGD$^+$ | HV | GAP | \|NDS\| |
| WS-LKH | 0.13 | 0.7022 | 0.06% | 73 | 0.10 | 0.7022 | 0.03% | 68 | 0.08 | **0.7161** | **0.00%** | 70 | 0.12 | 0.7019 | 0.07% | 66 |
| WS-ORTools | 0.37 | 0.6956 | 1.00% | 63 | 0.40 | 0.6942 | 1.17% | 63 | 0.38 | 0.7079 | 1.15% | 65 | 0.36 | 0.6958 | 0.94% | 65 |
| MOEA/D(T=5K) | 0.71 | 0.6897 | 1.84% | 119 | 0.50 | 0.6906 | 1.68% | 132 | 0.67 | 0.7016 | 2.02% | 113 | 0.64 | 0.6889 | 1.92% | 114 |
| NSGA-II(T=5K) | 1.91 | 0.6682 | 4.90% | 113 | 2.01 | 0.6608 | 5.92% | 105 | 2.03 | 0.6760 | 5.60% | 107 | 2.00 | 0.6621 | 5.74% | 107 |
| MOGLS(T=5K) | 1.89 | 0.6576 | 6.40% | 122 | 2.01 | 0.6536 | 6.95% | 101 | 2.09 | 0.6651 | 7.12% | 90 | 1.88 | 0.6588 | 6.21% | 108 |
| PPLS/D-C(T=200) | 1.10 | 0.6785 | 3.43% | 388 | 0.66 | 0.6879 | 2.06% | 519 | 0.64 | 0.7027 | 1.87% | 0.64 | 0.65 | 0.6881 | 2.04% | 546 |
| DRL-MOA | 3.69 | 0.6314 | 10.13% | 44 | 3.37 | 0.6336 | 9.79% | 45 | 3.41 | 0.6521 | 8.94% | 34 | 3.47 | 0.6364 | 9.40% | 39 |
| POMO-T | 0.56 | 0.6912 | 1.62% | 67 | 0.67 | 0.6886 | 1.96% | 64 | 0.63 | 0.7039 | 1.70% | 65 | 0.61 | 0.6900 | 1.77% | 69 |
| PMOCO | 0.56 | 0.6884 | 2.02% | 72 | 0.59 | 0.6883 | 2.01% | 73 | 0.61 | 0.7011 | 2.09% | 76 | 0.59 | 0.6887 | 1.95% | 77 |
| MLDRL | 0.58 | 0.6892 | 1.91% | 67 | 0.58 | 0.6895 | 1.84% | 74 | 0.58 | 0.7038 | 1.72% | 68 | 0.58 | 0.6902 | 1.74% | 77 |
| *MOEA/D+WRPN(T=5K)* | 0.35 | 0.6962 | 0.91% | 136 | 0.33 | 0.6958 | 0.94% | 146 | 0.32 | 0.7108 | 0.74% | 122 | 0.32 | 0.6961 | 0.90% | 146 |
| *NSGA-II+WRPN(T=5K)* | 0.17 | 0.7002 | 0.34% | 439 | 0.09 | 0.7011 | 0.19% | 381 | 0.17 | 0.7133 | 0.39% | 297 | 0.16 | 0.6991 | 0.47% | 373 |
| *MOGLS+WRPN(T=2K)* | 0.21 | 0.6992 | 0.48% | 193 | 0.23 | 0.6986 | 0.54% | 200 | 0.22 | 0.7129 | 0.45% | 186 | 0.23 | 0.6983 | 0.58% | 197 |
| *MOGLS+WRPN(T=5K)* | 0.14 | 0.7007 | 0.27% | 231 | 0.14 | 0.7003 | 0.30% | 224 | 0.15 | 0.7140 | 0.29% | 243 | 0.15 | 0.7002 | 0.31% | 250 |
| PMOCO(AUG) | 0.37 | 0.6938 | 1.25% | 73 | 0.37 | 0.6938 | 1.22% | 75 | 0.35 | 0.7079 | 1.15% | 72 | 0.39 | 0.6941 | 1.18% | 85 |
| MLDRL(AUG) | 0.37 | 0.6952 | 1.05% | 81 | 0.37 | 0.6946 | 1.11% | 75 | 0.32 | 0.7089 | 1.01% | 84 | 0.39 | 0.6944 | 1.14% | 82 |
| *MOGLS+WRPN(T=2K)(AUG)* | 0.08 | 0.7019 | 0.10% | 314 | 0.09 | 0.7015 | 0.13% | 376 | 0.09 | 0.7150 | 0.15% | 310 | 0.08 | 0.7016 | 0.11% | 343 |
| *MOGLS+WRPN(T=5K)(AUG)* | **0.04** | **0.7026** | **0.00%** | 382 | **0.05** | **0.7024** | **0.00%** | 413 | **0.05** | **0.7161** | **0.00%** | 360 | **0.04** | **0.7024** | **0.00%** | 399 |

### E.2    RESULTS OF DIFFERENT IMPLEMENTATIONS OF MOEAS

Considering that we implemented efficient parallelism MOEAs by ourselves based on their original frameworks, we further provide experimental results of the serial version of MOEAs based on the Pymoo library [2] (**MOEA/D (Pymoo)** and **NSGA-II (Pymoo)**) implementation with our replicated parallel version. It is worth mentioning that in the parallel version we implemented, we removed crossover operators, since the crossover operators hardly benefit the solution quality. Table 7 shows that the version we have implemented is superior on large-scale instances in terms of solution quality and running time.

---

[2] https://pymoo.org/

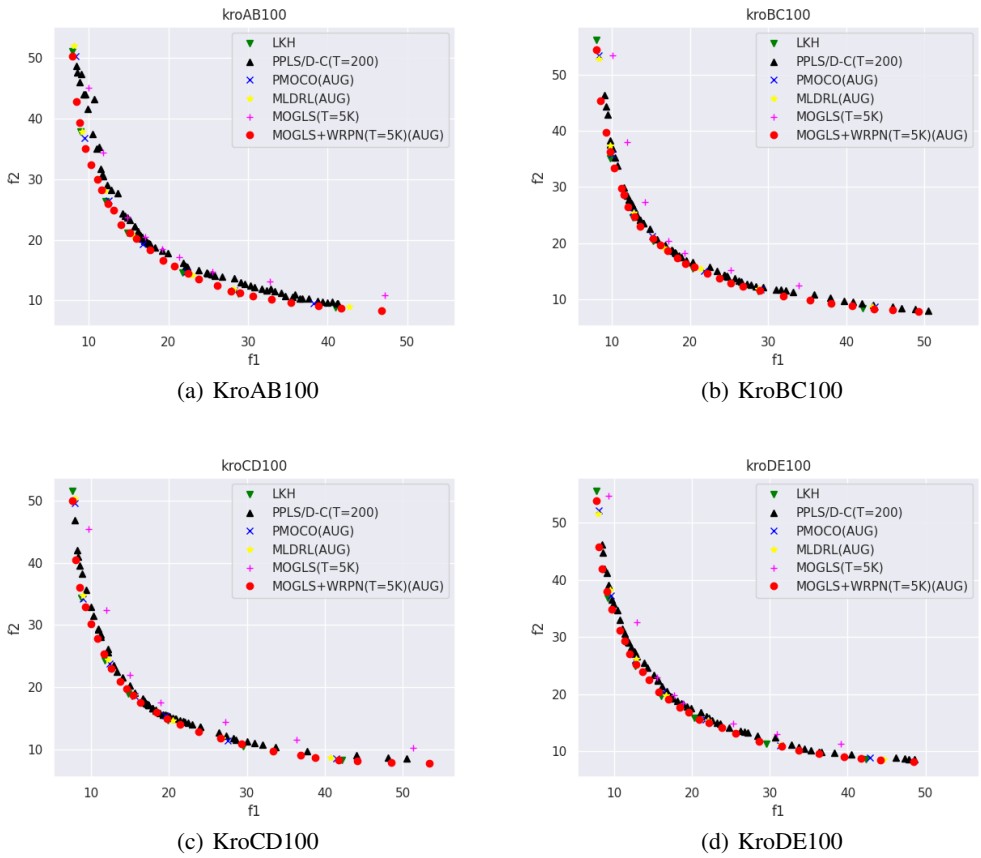

Figure 4: The Pareto front obtained on MOTSP benchmark instances. (Note: for clarity, we only put a dot for every 5 consecutive solutions along each PF.)

Table 7: Results of different implementations of MOEAs.

| Method | Bi-TSP-20 | | | | | Bi-TSP-50 | | | | | Bi-TSP-100 | | | | |
|---|---|---|---|---|---|---|---|---|---|---|---|---|---|---|---|
| | IGD$^+$ | HV | GAP | \|NDS\| | TIME | IGD$^+$ | HV | GAP | \|NDS\| | TIME | IGD$^+$ | HV | GAP | \|NDS\| | TIME |
| MOEA/D (Pymoo) | 0.15 | 0.6264 | 0.54% | 15 | 11m | 0.69 | 0.5994 | 7.08% | 82 | 14m | 3.74 | 0.5947 | 16.12% | 70 | 22.6m |
| NSGA-II (Pymoo) | **0.14** | **0.6294** | **0.06%** | 87 | 3.9m | **0.45** | 0.6248 | 3.15% | 100 | 8.5m | 3.88 | 0.6106 | 13.88% | 99 | 18.4m |
| MOEA/D | 0.35 | 0.6247 | 0.81% | 16 | 47s | 0.56 | **0.6336** | **1.78%** | 56 | 1.2m | **0.71** | **0.6951** | **1.96%** | 181 | 2.2m |
| NSGA-II | 0.17 | 0.6283 | 0.24% | 77 | 2.3m | 1.07 | 0.6145 | 4.74% | 98 | 2.3m | 2.09 | 0.6634 | 6.43% | 42 | 2.6m |

| Method | Bi-CVRP-20 | | | | | Bi-CVRP-50 | | | | | Bi-CVRP-100 | | | | |
|---|---|---|---|---|---|---|---|---|---|---|---|---|---|---|---|
| | IGD$^+$ | HV | GAP | \|NDS\| | TIME | IGD$^+$ | HV | GAP | \|NDS\| | TIME | IGD$^+$ | HV | GAP | \|NDS\| | TIME |
| MOEA/D (Pymoo) | 0.11 | 0.3991 | 7.38% | 6 | 4.9m | 0.86 | 0.3199 | 22.17% | 5 | 9.3m | 6.11 | 0.1979 | 51.26% | 4 | 16.6m |
| NSGA-II (Pymoo) | 0.09 | 0.4038 | 6.29% | 15 | 4.1m | **0.17** | 0.3785 | 7.91% | 9 | 9.2m | 3.47 | 0.2849 | 29.83% | 8 | 17.9m |
| MOEA/D | 0.10 | 0.4260 | 1.14% | 5 | 7m | 0.33 | **0.4012** | **2.87%** | 11 | 22.2m | **0.88** | **0.3976** | **2.07%** | 10 | 38.5m |
| NSGA-II | **0.06** | **0.4273** | **0.84%** | 14 | 5.7m | 0.75 | 0.3937 | 4.21% | 14 | 10.2m | 1.29 | 0.3821 | 5.89% | 13 | 14.7m |

| Method | Tri-TSP-20 | | | | | Tri-TSP-50 | | | | | Tri-TSP-100 | | | | |
|---|---|---|---|---|---|---|---|---|---|---|---|---|---|---|---|
| | IGD$^+$ | HV | GAP | \|NDS\| | TIME | IGD$^+$ | HV | GAP | \|NDS\| | TIME | IGD$^+$ | HV | GAP | \|NDS\| | TIME |
| MOEA/D (Pymoo) | 0.25 | 0.4588 | 3.94% | 86 | 12.9m | 1.83 | 0.3677 | 20.01% | 96 | 18m | 7.56 | 0.3284 | 37.21% | 95 | 27.4m |
| NSGA-II (Pymoo) | 0.79 | 0.4502 | 5.74% | 105 | 5.1m | 5.93 | 0.2746 | 40.27% | 105 | 10.8m | 19.13 | 0.1990 | 61.95% | 105 | 20.3m |
| MOEA/D | **0.23** | **0.4704** | **1.51%** | 104 | 48s | **0.61** | **0.4376** | **4.81%** | 627 | 78s | **2.19** | **0.4698** | **10.17%** | 978 | 3.3m |
| NSGA-II | 0.37 | 0.4572 | 4.27% | 517 | 2.2m | 3.55 | 0.3295 | 28.32% | 624 | 3.1m | 8.48 | 0.3301 | 36.88% | 588 | 3.4m |

## F  EFFECT OF DIFFERENT MODULES OF WRPN

### F.1  EFFECT OF OPERATORS

In our MOGLS+WRPN for MOTSP, the state transition for a solution executes the *ensemble* operator based on the action proposal. In Table 8, we evaluate the effectiveness of the *ensemble* operator against a single operator on MOTSP-50. The *2-opt* operator takes the longest running time but has the most significant effect among three single operators. It achieves similar performance as the *ensemble* operator when $T = 2K$ steps. However, it seems to trap into local optima and cannot get further improved when $T = 5K$ steps. In comparison, the *ensemble* operator makes the improvement process less vulnerable to falling into local optima.

Table 8: Comparison results on MOTSP-50 for different operators.

| Method | Operator | HV | GAP | Time |
|---|---|---|---|---|
| MOGLS+WRPN(T=2K) | *Exchange* | 0.6313 | 2.13% | 18s |
| | *Relocate* | 0.6409 | 0.64% | 17s |
| | *2-opt* | 0.6442 | 0.14% | 20s |
| | *Ensemble* | 0.6443 | 0.13% | 22s |
| MOGLS+WRPN(T=5K) | *Exchange* | 0.6350 | 1.57% | 43s |
| | *Relocate* | 0.6424 | 0.41% | 42s |
| | *2-opt* | 0.6441 | 0.15% | 52s |
| | *Ensemble* | **0.6451** | **0.00**% | 55s |

### F.2  EFFECT OF THE NUMBER OF NON-DOMINATED SOLUTIONS

Note that our WRPN may generate more than 100 non-dominated solutions. Its HV can take advantage of the mass number of solutions. Therefore, we design MOGLS+WRPN with truncated PF (**MOGLS+WRPN-**), which restricts the number of non-dominated solutions found by WRPN to an upper limit $100/105$ for Bi-TSP/Tri-TSP according to the crowding distance approach (Deb et al., 2002). Table 9 shows the comparison results. The good performance of MOGLS+WRPN- suffices to verify the quality of found solutions.

Table 9: Effect of the number of non-dominated solutions.

| Method | Bi-TSP-20 | | | | | Bi-TSP-50 | | | | | Bi-TSP-100 | | | | |
|---|---|---|---|---|---|---|---|---|---|---|---|---|---|---|---|
| | IGD$^+$ | HV | GAP | \|NDS\| | TIME | IGD$^+$ | HV | GAP | \|NDS\| | TIME | IGD$^+$ | HV | GAP | \|NDS\| | TIME |
| WS-LKH | 0.11 | 0.6270 | 0.44% | 15 | 3s | 0.14 | 0.6415 | 0.56% | 38 | 30s | 0.11 | **0.7090** | **0.00%** | 65 | 1.8m |
| *MOGLS+WRPN-(T=2K)* | 0.01 | 0.6296 | 0.03% | 76 | 15s | 0.09 | 0.6429 | 0.34% | 100 | 22s | 0.28 | 0.7044 | 0.65% | 100 | 44s |
| *MOGLS+WRPN-(T=5K)* | **0.00** | 0.6297 | 0.02% | 82 | 37s | 0.10 | 0.6426 | 0.39% | 100 | 55s | 0.24 | 0.7053 | 0.52% | 100 | 1.8m |
| *MOGLS+WRPN(T=2K)* | 0.01 | 0.6296 | 0.03% | 76 | 15s | 0.03 | 0.6443 | 0.12% | 255 | 22s | 0.22 | 0.7054 | 0.51% | 193 | 44s |
| *MOGLS+WRPN(T=5K)* | **0.00** | 0.6297 | 0.02% | 84 | 37s | 0.02 | 0.6446 | 0.08% | 327 | 55s | 0.15 | 0.7069 | 0.30% | 234 | 1.8m |
| *MOGLS+WRPN(T=5K)(AUG)* | **0.00** | **0.6298** | **0.00%** | 89 | 66s | **0.00** | **0.6451** | **0.00%** | 570 | 2.7m | **0.05** | **0.7090** | **0.00%** | 393 | 6.9m |

| Method | Tri-TSP-20 | | | | | Tri-TSP-50 | | | | | Tri-TSP-100 | | | | |
|---|---|---|---|---|---|---|---|---|---|---|---|---|---|---|---|
| | IGD$^+$ | HV | GAP | \|NDS\| | TIME | IGD$^+$ | HV | GAP | \|NDS\| | TIME | IGD$^+$ | HV | GAP | \|NDS\| | TIME |
| WS-LKH | 0.24 | 0.4712 | 1.34% | 61 | 4s | 0.51 | 0.4440 | 3.42% | 103 | 33s | 0.81 | 0.5076 | 2.94% | 105 | 2m |
| *MOGLS+WRPN-(T=2K)* | 0.23 | 0.4693 | 1.74% | 105 | 15s | 0.78 | 0.4321 | 6.03% | 105 | 26s | 1.58 | 0.4862 | 7.04% | 105 | 49s |
| *MOGLS+WRPN-(T=5K)* | 0.23 | 0.4687 | 1.86% | 105 | 40s | 0.76 | 0.4324 | 5.95% | 105 | 59s | 1.56 | 0.4874 | 6.81% | 105 | 2.2m |
| *MOGLS+WRPN(T=2K)* | 0.08 | 0.4759 | 0.36% | 486 | 15s | 0.23 | 0.4528 | 1.51% | 1269 | 26s | 0.54 | 0.5115 | 2.20% | 1436 | 49s |
| *MOGLS+WRPN(T=5K)* | 0.06 | 0.4766 | 0.21% | 693 | 40s | 0.16 | 0.4553 | 0.96% | 2190 | 59s | 0.36 | 0.5159 | 1.36% | 2478 | 2.2m |
| *MOGLS+WRPN(T=5K)(AUG)* | **0.01** | **0.4776** | **0.00%** | 1142 | 91s | **0.03** | **0.4597** | **0.00%** | 6971 | 3.6m | **0.06** | **0.5230** | **0.00%** | 8007 | 9.2m |

### F.3  EFFECT OF SOLVING TIME

Furthermore, we supplement the experiments in Table 10 by expanding the solving time of PMOCO and restricting the solving time of WRPN , but the results do not exhibit significant improvement with an increase in runtime cost.

Table 10: Effect of solving time.

| Method | Bi-TSP-20 | | | | | Bi-TSP-50 | | | | | Bi-TSP-100 | | | | |
|---|---|---|---|---|---|---|---|---|---|---|---|---|---|---|---|
| | $IGD^+$ | HV | GAP | \|NDS\| | TIME | $IGD^+$ | HV | GAP | \|NDS\| | TIME | $IGD^+$ | HV | GAP | \|NDS\| | TIME |
| WS-LKH | 0.23 | 0.6270 | 0.44% | 15 | 3s | 0.19 | 0.6415 | 0.56% | 38 | 30s | 0.12 | **0.7090** | **0.00%** | 65 | 1.8m |
| PMOCO | 0.20 | 0.6259 | 0.62% | 18 | 2s | 0.31 | 0.6351 | 1.55% | 49 | 4s | 0.54 | 0.6957 | 1.88% | 71 | 7s |
| MLDRL | 0.20 | 0.6271 | 0.43% | 26 | 2s | 0.28 | 0.6364 | 1.35% | 63 | 4s | 0.56 | 0.6969 | 1.71% | 73 | 7s |
| *MOGLS+WRPN(T=2K)* | **0.14** | 0.6296 | 0.03% | 76 | 15s | 0.09 | 0.6443 | 0.12% | 241 | 22s | 0.22 | 0.7055 | 0.49% | 192 | 44s |
| *MOGLS+WRPN(T=5K)* | **0.14** | 0.6297 | 0.02% | 84 | 37s | 0.08 | 0.6446 | 0.08% | 302 | 55s | 0.16 | 0.7069 | 0.30% | 233 | 1.8m |
| *MOGLS+WRPN(Restricted time)* | *0.14* | *0.6296* | *0.03%* | *83* | *2s* | *0.13* | *0.6428* | *0.36%* | *194* | *5s* | *0.34* | *0.7026* | *0.90%* | *221* | *12s* |
| PMOCO(AUG) | 0.23 | 0.6271 | 0.43% | 15 | 2s | 0.16 | 0.6401 | 0.78% | 51 | 5s | 0.33 | 0.7013 | 1.09% | 77 | 12s |
| PMOCO(AUG)(Extended time) | *0.22* | *0.6273* | *0.40%* | *17* | *55s* | *0.18* | *0.6407* | *0.73%* | *76* | *1.7m* | *0.26* | *0.7032* | *0.82%* | *169* | *3.7m* |
| *MOGLS+WRPN(AUG)(T=2K)* | **0.14** | **0.6298** | **0.00%** | 92 | 27s | 0.01 | 0.6450 | 0.02% | 490 | 60s | 0.10 | 0.7081 | 0.13% | 324 | 2.7m |
| *MOGLS+WRPN(AUG)(T=5K)* | **0.14** | **0.6298** | **0.00%** | 94 | 66s | **0.00** | **0.6451** | **0.00%** | 465 | 2.7m | **0.05** | **0.7090** | **0.00%** | 393 | 6.9m |

### F.4 Effect of the Number of Independent Executions

All learning-based methods employing the greedy strategy are executed only once, as their outcomes exhibit no randomness. For WS-LKH and WS-ORTools, we conduct a single run, because their results from 10 runs, despite consuming more runtime, are close to those from a solitary run, as discussed in Kool et al. (2019). For our WRPN and the baseline MOEAs, we have further performed 10 independent executions for Bi-TSP-20/50/100 and supplement the mean and standard deviations of these methods in HV, as presented in Table 11. The additional experiments indicated very minimal standard deviations, so for all experiments we run them only once to reduce time overheads.

Table 11: Effect of 10 independent executions on 200 randomly instances of Bi-TSP.

| Method | Bi-TSP-20 | | Bi-TSP-50 | | Bi-TSP-100 | |
|---|---|---|---|---|---|---|
| | mean | std. | mean | std. | mean | std. |
| MOEA/D | 0.6247 | $1.1 \times 10^{-4}$ | 0.6338 | $1.4 \times 10^{-4}$ | 0.6956 | $7.5 \times 10^{-5}$ |
| NSGA-II | 0.6284 | $9.1 \times 10^{-5}$ | 0.6147 | $1.1 \times 10^{-4}$ | 0.6708 | $6.2 \times 10^{-5}$ |
| MOGLS | 0.6286 | $4.1 \times 10^{-5}$ | 0.6296 | $9.9 \times 10^{-5}$ | 0.6504 | $1.6 \times 10^{-4}$ |
| *MOGLS+WRPN* | **0.6297** | $1.7 \times 10^{-6}$ | **0.6446** | $4.4 \times 10^{-6}$ | **0.7069** | $1.4 \times 10^{-5}$ |

## G Additional details of quality enhancement

In the quality enhancement process, several instances are first augmented. The instance augmentation for MOCOPs is presented in *construction* methods (Lin et al., 2022), which transform the original instance into multiple instances through a variety of efficient transformations, while all the instances share the same optimal solution. For a 2-dimensional coordinate $(x, y)$ in the space $[0, 1] \times [0, 1]$, it can be transformed by flipping or rotating. Typically, there are 8 common transformations, $\{(x, y), (y, x), (x, 1-y), (y, 1-x), (1-x, y), (1-y, x), (1-x, 1-y), (1-y, 1-x)\}$. When it is generalized to MOCOPs (e.g., MOTSP), there are $M$ 2-dimensional coordinates, and thus a total of $8M$ transformations can be obtained. In this paper, in order to reduce the inference time, we only consider one of $M$ 2-dimensional coordinates for a problem instance so as to obtain 8 transformed instances.

As an example, we provide the frameworks of a conventional MOGLS and its augmented L2I counterpart in Algorithm 3. The improvement process in MOGLS is modified into the L2I process in MOGLS_L2I. During the iterations, those non-dominated solutions are archived in the external population $EP_z$ for instance $\mathcal{I}_z$. In lines 13–15, some solutions may be discarded based on their crowding distances (Deb et al., 2002), if the external population has an upper limit. The final $EP$ is obtained by aggregating all $EP_z$ of different transformed instances.

The full results of our quality enhancement (QE) and vanilla instance augmentation (VIA) are presented in Table 12.

**Algorithm 3** Conventional MOGLS and its augmented L2I counterpart.

| | |
|---|---|
| **Input:** Instance set $\mathcal{I}$, weight vector distribution $\Lambda$, iteration number $T$. | **Input:** Instance set $\mathcal{I}$, policy network $\pi_\theta$, weight vector distribution $\Lambda$, iteration number $T$, number of transforms $Z$. |
| **Output:** External population $EP$. | **Output:** External population $EP$. |

| | |
|---|---|
| 1: **procedure** MOGLS($\mathcal{I}, \Lambda, T$) | 1: **procedure** MOGLS_L2I($\mathcal{I}, \Lambda, T, \pi_\theta, Z$) |
| 2:     (Empty) | 2:     $\{\mathcal{I}_1, \mathcal{I}_2, \cdots, \mathcal{I}_Z\} \leftarrow$ InstanceTransform($\mathcal{I}$); |
| 3:     $\delta \leftarrow$ InitialSolutions($\mathcal{I}$); | 3:     $\delta_z \leftarrow$ InitialSolutions($\mathcal{I}_z$), $\forall z \in \{1, \cdots, Z\}$; |
| 4:     $EP \leftarrow \varnothing$; | 4:     $EP, EP_z \leftarrow \varnothing, \forall z \in \{1, \cdots, Z\}$; |
| 5:     **for** $t \leftarrow 1 : T$ **do** | 5:     **for** $t \leftarrow 1 : T$ **do** |
| 6:         $\lambda \leftarrow$ SampleWeight($\Lambda$); | 6:         $\lambda \leftarrow$ SampleWeight($\Lambda$); |
| 7:         $\hat{\delta} \leftarrow$ Selection($\delta, \lambda$); | 7:         $\hat{\delta}_z \leftarrow$ Selection($\delta_z, \lambda$), $\forall z \in \{1, \cdots, Z\}$; |
| 8:         $o \leftarrow$ Crossover($\hat{\delta}$); | 8:         $o_z \leftarrow$ Crossover($\hat{\delta}_z$), $\forall z \in \{1, \cdots, Z\}$; |
| 9:         $o' \leftarrow$ **Improve**($o, \lambda$); | 9:         $o'_z \leftarrow$ **WRPN**($o_z, \lambda$), $\forall z \in \{1, \cdots, Z\}$; |
| 10:         $\delta \leftarrow$ Update($\delta, o'$); | 10:         $\delta_z \leftarrow$ Update($\delta_z, o'_z$), $\forall z \in \{1, \cdots, Z\}$; |
| 11:         $EP \leftarrow$ Archive($EP, o'$); | 11:         $EP_z \leftarrow$ Archive($EP_z, o'_z$), $\forall z \in \{1, \cdots, Z\}$; |
| 12:     **end for** | 12:     **end for** |
| 13:     (Empty) | 13:     **for** $z \leftarrow 1 : Z$ **do** |
| 14:     (Empty) | 14:         $EP \leftarrow$ Archive($EP_z, EP$); |
| 15:     (Empty) | 15:     **end for** |
| 16:     **return** $EP$; | 16:     **return** $EP$; |
| 17: **end procedure** | 17: **end procedure** |

Table 12: Experimental results of different enhancement mechanism on 200 randomly instances of Bi-TSP.

| Method | Bi-TSP-20 | | | | Bi-TSP-50 | | | | Bi-TSP-100 | | | |
|---|---|---|---|---|---|---|---|---|---|---|---|---|
| | HV | GAP | \|NDS\| | TIME | HV | GAP | \|NDS\| | TIME | HV | GAP | \|NDS\| | TIME |
| MOGLS+WRPN (VIA) (T=2K) | 0.6295 | 0.05% | 79 | 26s | 0.6425 | 0.39% | 182 | 56s | 0.7018 | 1.02% | 188 | 2.5m |
| MOGLS+WRPN (VIA) (T=5K) | 0.6296 | 0.03% | 92 | 62s | 0.6437 | 0.28% | 214 | 2m | 0.7044 | 0.65% | 228 | 5.7m |
| MOGLS+WRPN (QE) (T=2K) | **0.6298** | **0.00%** | 92 | 27s | 0.6450 | 0.02% | 490 | 60s | 0.7081 | 0.13% | 324 | 2.7m |
| MOGLS+WRPN (QE) (T=5K) | **0.6298** | **0.00%** | 94 | 66s | **0.6451** | **0.00%** | 465 | 2.7m | **0.7090** | **0.00%** | 393 | 6.9m |

## H COMPARISON ON SINGLE OBJECTIVE WITH DACT

While the core idea of our method and single-objective L2I is to achieve approximate optimal solutions through improvement, our method differs significantly from single-objective L2I in terms of network structure, training algorithm, action space, etc. Our WRPN approach enables the parallel improvement of multiple solutions and can be easily integrated with the general MOEA framework. In addition to the MOCOPs, we adapted WRPN to solve SOCOPs by fixing weights in a specific direction for the search. Table 13 presents WRPN on VRP-20 and VRP-50 with the objective solely on optimizing the first objective $f_1$ (i.e., the total routing cost) in Bi-VRP. Although WRPN performs slightly inferior to DACT, it still achieves competitive results.

Table 13: Experimental results of WRPN and DACT on 200 randomly instances of single-objective VRP.

| Method | VRP-20 | | | VRP-50 | | |
|---|---|---|---|---|---|---|
| | $f_1$ | GAP | TIME | $f_1$ | GAP | TIME |
| DACT(T=2K) | 6.105 | 0.03% | 1.3m | 10.342 | 0.17% | 2.2m |
| DACT(T=5K) | **6.099** | **-0.08%** | 3.7m | **10.282** | **-0.40%** | 6.2m |
| *WRPN(T=2K)* | 6.108 | 0.08% | 1.3m | 10.374 | 0.49% | 2.2m |
| *WRPN(T=5K)* | 6.104 | 0.00% | 3.6m | 10.324 | 0.00% | 6.2m |

## I  MODEL HYPERPARAMETERS

In addition to the various parameters mentioned in the training details in Section 5.1. More detailed hyperparameters settings of our WRPN are listed in Table 14.

Table 14: List of model hyperparameters.

| Parameter | Value |
|---|---|
| Number of stacked encoders $L$ | 3 |
| Number of attention heads $m$ | 4 |
| Embedding dim | 64 |
| Hidden dim | 64 |

## J  USED ASSETS AND LICENSES

Table 15 lists the assets used in our work, which are all open-source for academic research. We use the MIT license for our code and the used data (new assets).

Table 15: Used assets and their licenses.

| Type | Asset | License | Usage |
|---|---|---|---|
| Code | LKH (Tinós et al., 2018) | Available for academic use | Evaluation |
|  | ORTools Tinós et al. (2018) | Available for academic use | Evaluation |
|  | NSGA-II (Deb et al., 2002) | MIT License | Remodification and evaluation |
|  | MOEA/D (Ke et al., 2013) | MIT License | Remodification and evaluation |
|  | MOGLS (Jaszkiewicz, 2002) | MIT License | Remodification and evaluation |
|  | PPLS-D/C (Shi et al., 2022) | MIT License | Remodification and evaluation |
|  | DRL-MOA (Li et al., 2020) | MIT License | Remodification and evaluation |
|  | POMO-T (Kwon et al., 2020) | MIT License | Remodification and evaluation |
|  | PMOCO (Lin et al., 2022) | MIT License | Remodification and evaluation |
|  | MLDRL (Zhang et al., 2022) | MIT License | Remodification and evaluation |
| Datasets | TSPLIB | Available for any non-commerial use | Testing |

