# OpenReview forum: "Solving Multiobjective Combinatorial Optimization via Learn to Improve Method"
_ICLR.cc/2024/Conference — Submitted to ICLR 2024_

### Official Review · Reviewer_ru1d · 2023-10-28

**Soundness:** 4 excellent
**Presentation:** 2 fair
**Contribution:** 2 fair
**Rating:** 6
**Confidence:** 4

**Summary:**

One of the traditional methods for solving MOCOPs involves the MOEA approach. In this approach, individual solutions in a population pool are continuously updated through cross-overs and enhanced via local search. The authors suggest replacing the local search component within the MOEA. Previously, the local search was driven by a random selection of node pairs; however, the authors now employ a neural network for this pair selection.

**Strengths:**

1. This is the first L2I approach applied to MOCOPs.

2. A new neural net architecture is introduced, which can accommodate the weight factor used in identifying the Pareto set.

3. A new RL training method is introduced, leveraging the population pool of the MOEA approach.

4. A new quality enhancement method suitable for the MOEA approach is presented.

**Weaknesses:**

While the authors have commendably applied the L2I approach to MOCOPs, yielding impressive results, the novelty of the ideas underpinning this work doesn't fully meet the expectations I have for ICRL publications.

1. The concept of utilizing a trained neural model to bolster the local search component of a genetic algorithm isn't novel.

2. The presented neural net architecture appears to be a minor variation of an existing one, specifically DACT.

3. The employment of a shared baseline for REINFORCE isn't groundbreaking, as seen in POMO.

4. Similarly, quality enhancement via instance augmentation isn't a pioneering approach.

While I acknowledge that the specific methodologies deployed in the paper are novel, especially given this is the inaugural L2I application to MOCOPs, the broader insights readers can derive from this work seem somewhat limited.

**Questions:**

Branding the methodology in this paper as "L2I" might lead to confusion in the future. As subsequent research emerges that applies the L2I approach to MOCOPs, referencing this work simply as "L2I" could create ambiguity. Future works that aim to compare their results with this paper would face challenges in distinguishing between this specific approach and other "L2I" methodologies for MOCOPs.

---

> ### Author Response · Authors · 2023-11-20
>
> Thank you for your valuable comments.
>
> While MOCOPs can be decomposed into multiple scalar optimization subproblems and solved individually using existing local improvement algorithms, this approach often overlooks the dominance relationships among Pareto solution sets. Each solution is improved independently, which can result in neighboring subproblems converging to similar solutions and a loss of diversity. Furthermore, existing local improvement algorithms require separate sub-models to be trained for each subproblem.
>
> To illustrate the difference, consider the table below, which compares the number of model parameters required by the classical single-objective local improvement algorithm DACT and our L2I when solving multi-objective problems (e.g. Bi-TSP decomposed into 101 subproblems). It is evident that DACT requires a significantly larger number of model parameters compared to L2I, resulting in a more challenging model to train and longer training time.
>
> | Method | #(model) | #(parameter) |
> |-|-|-|
> | DACT | 101 | 30B|
> | L2I | 1 | 0.298B |
> ||||
>
> Our L2I approach offers a more flexible and efficient neural network that can solve multiple subproblems simultaneously and improve their solutions collaboratively. This mitigates the issue of a lack of diversity in the Pareto solution set. In addition, L2I incorporates a shared baseline and quality enhancement mechanism, which captures implicit connections among Pareto solutions during both the training and inference processes. The ablation experiments in Section 5.3 further validate the effectiveness of our approach.
>
> We further respond to your concerns as follows.
>
> **To W1:**
>
> Although there are similar methods previously proposed for single-objective combinatorial optimization problems, it is challenging to modify these methods for MOCOPs. We present the initial attempt of using the learn-to-improve framework. It supports collaboratively improvement of multiple solutions and has achieved promising results.
>
> **To W2:**
>
> DACT is one of the learn-to-improve methods for single-objective COPs, which cannot directly address MOCOPs. In principle, any advanced neural model can be aggregated in our weighted-related policy network, where DACT is just a building block. When directly training multiple DACT models to deal with the decomposed subproblems for MOCOPs, it costs more computational resources and exhibits poor performance compared with our model, as shown in Table 3.
>
> **To W3:**
>
> The shared baseline in L2I is derived by solving multiple subproblems for a MOCOP simultaneously and averaging their solutions. On the other hand, the shared baseline in POMO is obtained by solving a single-objective problem multiple times and averaging the results. However, applying POMO strategy to the multi-objective domain is challenging, because individually solving each scalar subproblem multiple times would consumes a significantly increasing amount of memory and runtime, making it impractical.
>
> **To W4:**
>
> The vanilla instance augmentation approach for single-objective COPs exploits equivalent instance transformation, which could be directly applied to the decomposed subproblems for MOCOPs, as in [1]. Different from the vanilla instance augmentation approach, our quality enhancement approach is specifically tailored for MOCOPs, further utilizing the external population of MOEAs and the Pareto dominance of solutions. The comparison results in Table 4 show our quality enhancement approach can better improve the solution quality compared with the vanilla instance augmentation approach.
>
> [1] Xi Lin, Zhiyuan Yang, and Qingfu Zhang. Pareto set learning for neural multi-objective combinatorial optimiza- tion. In International Conference on Learning Representations, 2022.
>
>
>
> **To Questions:**
>
> Thank you for your suggestion. We have taken it into consideration and decided to revise the name of our model to Weighted-Related Policy Network (WRPN). This change aims to prevent potential confusion with other L2I approaches.

---

> > ### Comment · Reviewer_ru1d · 2023-11-20
> >
> > Thank you for your clarification. I have changed my score.

---

> > > ### Author Response · Authors · 2023-11-20
> > >
> > > We appreciate the reviewer for acknowledging our response and supporting our work.

---

### Official Review · Reviewer_Rq8H · 2023-10-31

**Soundness:** 3 good
**Presentation:** 2 fair
**Contribution:** 3 good
**Rating:** 8
**Confidence:** 3

**Summary:**

This paper presents a new deep reinforcement learning method that involves a learning-based improvement method for solving multi-objective combinatorial optimization problems. A weight-related policy network is embedded into multi-objective evolutionary algorithm frameworks to guide the search. Experimental studies on multi-objective traveling salesman problems and multi-objective vehicle routing problems show the effectiveness of the proposed method.

**Strengths:**

1.	A learning-based improvement method is proposed for solving multi-objective combinatorial optimization problems with deep reinforcement learning.
2.	An ablation study is conducted to study the proposed method and show its effectiveness.

**Weaknesses:**

1.	The way the proposed weight-related policy network is embedded is not clearly described. A pseudo-code of the complete method for solving one multi-objective combinatorial optimization problem should be provided.
2.	The proposed method contains several hyperparameters, e.g., the number of transformer-style stacked encoders and the number of attention heads. A summary of them needs to be provided.
3.	How many times does each algorithm run independently in the experiment?

**Questions:**

1.	How does the algorithm perform when using complex operators?
2.	What are the numbers of variables for the test problems in the experiments?

---

> ### Author Response · Authors · 2023-11-20
>
> **To W1**:
>
> Thanks for your suggestions. Initially, we presented a detailed explanation of the proposed weight-related policy network in Sections 4.2 and a pseudo-code outlining the L2I inference process in Algorithm 2 in Appendix G. In response to your feedback, we will include pseudo-code detailing the computation of the weight-related policy network when addressing an MOCOP with L2I, as demonstrated in Algorithm 2.
>
> **To W2**:
>
> Thank you for your suggestion. We will include a list of hyperparameters in Appendix I, aiming to enhance the understanding of the model's specific parameters.
>
> **To W3**:
>
> All learning-based methods employing the greedy strategy are executed only once, as their outcomes exhibit no randomness. For WS-LKH and WS-ORTools, we conduct a single run, because their results from 10 runs, despite consuming more runtime, are close to those from a solitary run, as discussed in [1].
>
> For our L2I and the baseline MOEAs, we also run them only once, as the standard deviations are tiny. To show this, we have further supplemented the results with 10 independent executions for Bi-TSP-20/50/100, as presented in Table 11.
>
> | Method | Bi-TSP-20 | | Bi-TSP-50 | | Bi-TSP-100 | |
> |-|-|-|-|-|-|-|
> | | mean |std. |  mean | std.  | mean | std. |
> | MOEA/D | 0.6247 | 1.1$\times 10^{-4}$  | 0.6338 | 1.4$\times 10^{-4}$ |  0.6956 | 7.5$\times 10^{-5}$ |
> | NSGA-II | 0.6284 | 9.1$\times 10^{-5}$ | 0.6147 | 1.1$\times 10^{-4}$ | 0.6708 | 6.2$\times 10^{-5}$ |
> | MOGLS | 0.6286 | 4.1$\times 10^{-5}$ | 0.6296 | 9.9$\times 10^{-5}$ | 0.6504 | 1.6$\times 10^{-4}$  |
> | MOGLS+L2I | **0.6297** | 1.7$\times 10^{-6}$ | **0.6446** | 4.4$\times 10^{-6}$ | **0.7069** | 1.4$\times 10^{-5}$ |
> ||||||||
>
> [1] Wouter Kool, Herke van Hoof, and Max Welling. Attention, learn to solve routing problems! In International Conference on Learning Representations, 2019.
>
>
> **To Q1:**
>
> It is possible to employ more complex operators to further enhance the search process, but this may require additional computational resources during both training and inference. In our methodology, we utilize an ensemble operator consisting of three basic ones. The results of the ablation study are presented in Table 8 in Appendix F.1. It is evident from the results that the ensemble operator, being a more complex operator, surpasses the performance of three basic operators.
>
> **To Q2:**
>
> The number of decision variables is closely related to the length of the decision sequence. In the case of MOTSP, the number of decision variables equals the number of nodes. In our paper, we consider three specific numbers: 20, 50, and 100. For MOCVRP, additional dummy depots are included in the decision sequence, as depicted in Appendix B.2. In our analysis, we evaluate three sizes of decision variables: 30, 70, and 120, corresponding to 20, 50, and 100 nodes, respectively.

---

### Official Review · Reviewer_BJaC · 2023-11-01

**Soundness:** 4 excellent
**Presentation:** 4 excellent
**Contribution:** 3 good
**Rating:** 8
**Confidence:** 4

**Summary:**

The paper introduces "Learn to Improve" (L2I), a deep reinforcement learning (DRL) technique designed to address multiobjective combinatorial optimization problems (MOCOPs). L2I contrasts traditional DRL methods by embedding a weight-related policy network into multiobjective evolutionary algorithm (MOEA) frameworks. This assists in directing the search, reduces training variance, and offers an enhanced quality mechanism for better model inference. Computational experiments on classic MOCOPs like multiobjective traveling salesman and vehicle routing problems highlight the superiority of L2I over existing methods.

**Strengths:**

L2I introduces a new DRL-based approach for MOCOPs. Unlike the traditional "Learn to Construct" methodology, L2I emphasizes iterative improvements. The L2I module has demonstrated adaptability as it can be integrated into various MOEA frameworks, such as NSGA-II, MOEA/D, and MOGLS. This mechanism, applying instance augmentation techniques, improves both the proximity and diversity of the Pareto set. The L2I methodology outperforms other state-of-the-art techniques on standard MOCOPs, even showing better performance than renowned solutions like the LKH solver for specific problems.

**Weaknesses:**

Since the paper deals with the combinatorial optimization, decomposition methods are not sufficiently elaborated/reviewed.

**Questions:**

How does L2I compare with methods follows a general scheme of "learn-divide-and-conquer" or "divide-learn-and-conquer"? In a sense, there are approaches that learn how to decompose a problem and there are approaches that decompose a problem before learning. I qualitative assessment may be sufficient.

---

> ### Author Response · Authors · 2023-11-20
>
> Thank you for your valuable comments.
>
> Many multiobjective algorithms first decompose the MOCOP into a number of scalar optimization subproblems, and then optimize them simultaneously. To the best of our knowledge, approaches that learn how to decompose a problem are rare. Exploring this concept could present an interesting direction for further investigation. On the other hand, our L2I approach focuses on learning improvement strategies for each individual subproblem. It belongs to the scheme of “divide-learn-and-conquer”.

---

### Official Review · Reviewer_P9jg · 2023-11-02

**Soundness:** 3 good
**Presentation:** 2 fair
**Contribution:** 3 good
**Rating:** 8
**Confidence:** 4

**Summary:**

This paper proposes a new paradigm "learn to improve" in the context of solving Mult objective combinatorial optimization problems (MOCOPs). The proposed approach adds an improvement operation, based on a deep policy network, that works in parallel with individual solutions, using evolutionary technique. The deep network is based on an Encoder-Decoder, where encoder is Transformer-style stacked encoders, with Dual-Aspect Collaborative Attention (DAC-Att). Comparisons with SOA and ablation studies are done.

**Strengths:**

a. Mathematical fomulation of the proposed method
b. Details of the proposed deep network based on Encoder-Decoder based on Transformer.
c. The use of Dual-Aspect Collaborative Attention (DAC-Att)
d. Details of the algorithm
e. Comparisons with state of the art (SOA) on MOCOPs.
f. Ablation Study

**Weaknesses:**

a. In the result tables, the proposed method is not highlighted
b. The results in tables 1 & 2 are not discussed on why the proposed approach is better only in the last entries?

**Questions:**

Why the proposed method show better results? Please explain the specificality of the proposed method.

---

> ### Author Response · Authors · 2023-11-12
>
> Dear Reviewer P9jg,
>
> Thanks for the detailed review.
>
> It looks like the review is about another paper. E.g., It emphasizes a fault prediction method, but it is not mentioned in our paper. We would appreciate it if you could check it and offer an appropriate review.
>
> Best,
>
> Authors

---

> > ### Comment · Reviewer_P9jg · 2023-11-21
> > **MISTAKE CORRECTION**
> >
> > I accidentally entered the wrong (another paper) review. Now I have corrected it. I sincerely apologize for this mistake.

---

> ### Author Response · Authors · 2023-11-22
>
> Thanks for your updates and valuable comments.
>
> **To Weakness a:**
>
> According to your suggestions, we have highlighted our proposed method in italic in the tables.
>
> **To Weakness b:**
>
> In terms of the improved outcomes reported in the tables, there are two points to clarify.
>
> Firstly, various MOEA frameworks, when augmented with our L2I approach, retain their distinct original characteristics. Notably, MOGLS, designed specifically for multi-objective combinatorial optimization, achieves the best result. However, MOEA/D and NSGA-II, originally proposed for general multi-objective optimization, exhibit comparatively lower performance.
>
> Secondly, one of the results obtained from MOGLS+L2I in the last entry benefits from a larger number of iterations and the incorporation of a quality enhancement mechanism. As a result, this particular configuration demonstrates the best performance.
>
> **To Q:**
>
> On the one hand, our L2I approach differs from traditional MOEAs as it leverages deep reinforcement learning techniques. Through training on extensive datasets, L2I acquires the ability to selectively invoke local operators that enhance solutions in a specified direction.
>
> On the other hand, L2C methods directly construct solutions for individual subproblems, which can lead to a diversity problem due to the inadequate search for potential Pareto solutions in the neighborhood of each subproblem. In contrast, our L2I approach adopts a different learning paradigm. It collaboratively improves multiple solutions, effectively addressing the aforementioned issues and enhancing the solution quality.

---

### Author Response · Authors · 2023-11-20
**General Response: Revision Summary**

We thank all reviewers for their valuable and constructive suggestions, which greatly contributed to the improvement of our paper.   In response, we have conducted additional experiments to address the raised concerns. We summarize these modifications below:

- We provide the complete pseudo-code for L2I solving an MOCOP in Appendix B.

- We include the ablation study the number of independent executions of MOEAs in Appendix F.4.

- We provide additional hyperparameters for the model in Appendix I.

- We modified the expression of L2I and revise the name of our model to Weighted Relevant Policy Network (WRPN), according to the suggestion of Reviewer ru1d.

All revisions in the PDF file have been highlighted in red.

---

### Meta-Review · Area_Chair_KzZi · 2023-12-06

**Metareview:**

This paper introduces a learning-to-improve (L2I) method on the multi-objective combinatorial optimization problems (MOCOPs). Specifically, it adapts the L2I method named DACT  (a transformer-style policy network) from single-objective settings of COPs to a multi-objective settings, and replaces the local search component in the multi-objective evolutionary algorithm (MOEA) with the adapted DACT. Experiments on multi-objective Traveling Salesman Problem and vehicle routing problems demonstrate the enhancement of MOEA using the DACT-like model for iterative improvement.

On the positive side, the reviewers (BJaC, ru1d and the AC) recognize this work as the first attempt of using a DACT-like model for L2I within a multi-objective formulation of COPs (combinatorial opt. prob.), which is a meaningful adaptation of the original DACT for single-objective COPs. The L2I module has demonstrated its adaptability (BJaC) as it can be integrated into the MOEA framework and similar ones, which means a broad range of representative classical heuristic methods for MOCOPs could be benefited from such an integration.  This makes the work a valuable contribution in MOCOPs.
It also has a careful ablation study in verifying the effectiveness of the proposed component (P9jg, Rq8H).

On the negative side, this paper includes a large portion of the formula, training algorithm, reinforcement learning formulation (state space, reward function, etc.) being identical or highly similar to those in the original DACT.  However, the paper  does  not make it explicit which parts are directly from the original DACT, and which parts are indeed novel in this work.  This is not acceptable as it does not portray the right picture, potentially misleading in novelty claim/assessment.   In addition, the paper claims to be the SOTA method but without comparison with other strong heuristic methods [1,2,3]. The authors mistakenly equated the SOTA machine-learning-based method as the SOTA method on MOCOPs.

To conclude, this paper should address the credibility issue by clarifying their novelty claim and rewriting the method description sections, making it explicit which parts are from previous work and which parts are new in this work, and by extending the experiments with more strong baselines (including the non-learning based strong methods).  If such additional experiments are not possible due to time constraints, the authors should at least modify their claimed advantages within the correct scope of their experiments. The current version of the paper cannot be accepted for publication.

**Justification For Why Not Higher Score:**

Issues in the clarity on novelty claim and in the scope of empirical comparison.

**Justification For Why Not Lower Score:**

Demonstrating  the enhancement by a DRL module in frameworks of heuristic methods is a valuable contribution in MOCOPs.

---

### Decision · Program_Chairs · 2024-01-16

Reject